# Aggregation induced emission dynamic chiral europium(III) complexes with excellent circularly polarized luminescence and smart sensors

Yun-Lan Li[1], Hai-Ling Wang[1], Zhong-Hong Zhu [1] ✉, Yu-Feng Wang[1], Fu-Pei Liang [1] ✉ & Hua-Hong Zou [1] ✉

The synthesis of dynamic chiral lanthanide complex emitters has always been difficult. Herein, we report three pairs of dynamic chiral $Eu^{III}$ complex emitters (**R/S-Eu-R-1**, R = Et/Me; **R/S-Eu-Et-2**) with aggregation-induced emission. In the molecular state, these $Eu^{III}$ complexes have almost no obvious emission, while in the aggregate state, they greatly enhance the $Eu^{III}$ emission through restriction of intramolecular rotation and restriction of intramolecular vibration. The asymmetry factor and the circularly polarized luminescence brightness are as high as 0.64 ($^5D_0 \rightarrow {}^7F_1$) and 2429 $M^{-1}cm^{-1}$ of **R-Eu-Et-1**, achieving a rare double improvement. **R-Eu-Et-1/2** exhibit excellent sensing properties for low concentrations of $Cu^{II}$ ions, and their detection limits are as low as 2.55 and 4.44 nM, respectively. Dynamic $Eu^{III}$ complexes are constructed by using chiral ligands with rotor structures or vibration units, an approach that opens a door for the construction of dynamic chiral luminescent materials.

Lanthanide complex emitters with many advantages such as narrow emission peak, good monochromaticity, large Stokes shift, long excited state lifetime, and strong light absorption ability have attracted great attention[1–4]. The complex electronic configuration of the $4f^n$ electron shell of trivalent lanthanide metal ions lead to electron excitation to the high-energy $4f$ orbital which is hardly affected by the ligands in complexes. Therefore, lanthanide complex emitters have bright and clear emission lines[5,6]. Rational construction of lanthanide complex emitters with excellent luminescent properties has promoted the rapid development of smart sensing, solid-state lighting, anti-counterfeiting, and biodiagnostics[7–10]. The short triplet lifetimes of most organic ligands and their poor energy-level matching with lanthanide ions hinder the efficient construction of lanthanide complex emitters[11,12]. Although a variety of organic ligands have been synthesized to act as antennas to solve the above problems, the energy levels of organic ligands with specific structures and fixed configurations are precise, and effectively constructing dynamic lanthanide complex

emitters is impossible[13,14]. Therefore, reconsidering the method of synthesizing lanthanide complexes with efficient dynamic luminescence properties is urgently needed.

In 2001, Tang et al. discovered that the fluorescence of organic fluorophores with a molecular rotor structure gradually rises with the increase of the degree of aggregation, and they proposed aggregation-induced emission (AIE)[15]. Currently, restriction of intramolecular motion (RIM) is the most recognized mechanism for constructing efficient and bright AIE fluorophores (AIEgens), and this method can be further divided into the restriction of intramolecular rotation (RIR) and restriction of intramolecular vibration (RIV) mechanisms[16,17]. In recent years, some dynamic luminescent complexes which rely on the RIM mechanism and exhibit AIE behavior have gradually appeared and have shown attractive application prospects in the fields of intelligent sensing, multiple anti-counterfeiting, solid-state lighting, circularly polarized luminescence, and biological diagnosis and treatment[18–22]. In 2020, Zang et al. used the chiral ligand (R/S)−2,2′-bis(di-p-

[1]School of Chemistry and Pharmaceutical Sciences, State Key Laboratory for Chemistry and Molecular Engineering of Medicinal Resources, Guangxi Normal University, Guilin 541004, P. R. China. ✉e-mail: 18317725515@163.com; liangfupei@glut.edu.cn; gxnuchem@foxmail.com

tolylphosphine)−1,1′-binaphthalene to coordinate with the $Cu^I$ ion to obtain a pair of chiral $Cu^I$ complex isomers which show photoresponse performance and circularly polarized luminescence (CPL) performance[23]. Next, the research group used a butterfly-like organic ligand oxacalix[2]arene[2]pyrazine to construct a dynamic luminescent metal-organic framework (MOF) with RIV properties through a simple solvothermal reaction[24]. Although some progress has been made in the rational design and synthesis of dynamic luminescent MOFs and transition metal complex emitters, the construction of dynamic lanthanide complex emitters has not yet been realized. Selecting organic ligands with obvious molecular rotor structures or vibrational units which match the energy levels of $Ln^{III}$ ions is one of the most effective ways to rationally construct dynamic lanthanide complex emitters.

In recent years, chiral complex emitters have become a research hotspot and have potential applications in the fields of nonlinear optics, anti-counterfeiting inks, three-dimensional displays, and polarized light microscopy[25–29]. Thus far, achieving both high luminescence asymmetry factor ($g_{lum}$) and circularly polarized luminescence brightness ($B_{CPL}$) is difficult for most chiral luminescent materials[30]. High $B_{CPL}$ values can depend on molecules with excellent molar absorptivity, but their relatively low $g_{lum}$; however, unusually strong $g_{lum}$ can be hampered by weak molar absorptivity and quantum yield (QY), resulting in unsatisfactory $B_{CPL}$ values. Chiral lanthanide complexes usually exhibit high $g_{lum}$ values given the magnetic dipole-allowed transitions of lanthanide metal ions[31–33]. In addition, the characteristic radiative transitions of lanthanide emitters can be promoted by enhancing the antenna effect to obtain high QY[34–36]. Therefore, chiral lanthanide complex emitters are one of the best candidates for constructing CPL materials with high-performance parameters, and they are expected to achieve double improvement in $g_{lum}$ and $B_{CPL}$ values.

Heavy metal ion pollution has become a serious environmental problem. Such pollutants often exist in industrial wastewater and cannot be degraded by organisms, eventually entering the food chain system, and seriously threatening human health and the balance of natural ecosystems. Especially for the heavy metal ion $Cu^{II}$, excessive absorption by plants can cause physiological metabolic disorders, and growth and development retardation poisoning; further, a large amount of $Cu^{II}$ can combine with albumin in serum to cause damage to the liver and central nervous system[37,38]. The United States Environmental Protection Agency (USEPA) prescribes that the maximum permission concentration of $Cu^{II}$ ions in drinking water cannot exceed $20\,\mu M$[14]. Sensitivity is one of the most important metrics for measuring high-performance sensors[39,40]. Therefore, the development of highly sensitive sensors capable of fast and intelligent responses to very low concentrations of $Cu^{II}$ ions is of great importance.

Herein, we used chiral organic ligands with molecular rotor structures or vibrational units to react with lanthanide metal ions under solvothermal conditions and obtained three pairs of dynamic chiral $Eu^{III}$ complex isomers with RIM characteristics (***R/S*-Eu-R-1** (R = Et/Me) and ***R/S*-Eu-Et-2**). The simple, efficient, high-yield, and atom-utilizing solvothermal one-pot method effectively avoids the complicated organic synthesis process and provides a strategy for the rational construction of dynamic lanthanide complex emitters. Note that with the gradual increase of glycerin, the emission intensity of the four chiral $Eu^{III}$ complex emitters increases significantly, showing $\alpha_{AIE}$ values as high as 92.54/87.95 (***R/S*-Eu-Et-1**), and 13.44/16.8 (***R/S*-Eu-Et-2**). In addition, the chiral $Eu^{III}$ complex ***R*-Eu-Et-1** has both high $g_{lum}$ (0.64) and $B_{CPL}$ (2429 $M^{-1}cm^{-1}$), realizing the double improvement of CPL parameters. Both ***R*-Eu-Et-1/2** in the aggregated state exhibit highly sensitive photoresponses to very low concentrations of $Cu^{II}$ ions, and their detection limits are as low as 2.55 and 4.44 nM, respectively, much lower than the maximum permissible value (20 μM) of $Cu^{II}$ ion in drinking water stipulated by USEPA. High-resolution electrospray mass spectrometry (HRESI-MS) results demonstrate that $Cu^{II}$ ions replace

$Ln^{III}$ ions in the chiral lanthanide complex emitters to induce strong ligand-to-metal charge transfer (LMCT) leading to luminescence quenching. Furthermore, ***R*-Eu-Et-1** exhibits excellent photoresponses to both low-concentration acidic and alkaline aqueous solutions. This work synthesized a series of chiral dynamic lanthanide complex emitters and developed a strategy for the rational design and construction of dynamic lanthanide complex emitters (Fig. 1). In addition, this research provides a perspective for the design and synthesis of metal complex emitters with CPL performance and multiple sensing.

## Results

### Crystal structural analysis of *R/S*-Eu-R-1 (R = Et/Me), *R/S*-Eu-Et-2, and Eu-Et-3

Accurately weighed $(1R/S,2R/S)$-(-/ + )−1,2-diphenylethylenediamine, 1-methyl/ethyl-2-imidazolecarboxaldehyde, and $Eu(NO_3)_3\cdot6H_2O$ with a stoichiometric ratio of 1:1:1 were dissolved in a mixed solvent of EtOH and $CH_3CN$ (1:1) and underwent a solvothermal reaction under one-pot conditions, an approach which easily and quickly generated two pairs of chiral $Eu^{III}$ complexes ***R/S*-Eu-R-1** (R = Et/Me) containing molecular rotor structures in high yield. The one-pot synthesis strategy under solvothermal conditions effectively avoids tedious organic synthesis, improves atom utilization, and reduces cost and time. Single crystal X-ray diffraction (SCXRD) results show that ***R/S*-Eu-R-1** is crystallized in the orthorhombic chiral space group $P2_12_12_1$ (Supplementary Table 1). ***R/S*-Eu-R-1** (R = Et/Me) are enantiomers of each other, and they have similar structural connections (Supplementary Fig. 1A). As shown in Figs. 2A, R-Eu-Et-1 consists of a $Eu^{III}$ ion, a chiral Schiff base ligand (1E,1′E)-N,N′-(1,2-diphenylethane-1,2-diyl)bis(1-(1-ethyl-1H-imidazol-2-yl)methanimine) (*R*-L[1]) with multidentate chelate coordination and molecular rotors, and the combination of three end-group coordinated $NO_3^-$ ions, and its molecular formula is $[Eu(R-L^1)(NO_3)_3]$. In the molecular state, the ligand *R*-L[1] in the structure of ***R*-Eu-Et-1** has two molecular rotors that can rotate freely. In the aggregated state, the molecular rotors between two adjacent ***R*-Eu-Et-1** molecules cross-stack each other and greatly restrict the free rotation of the benzene ring rotors through strong hydrogen bonds and steric hindrance (Fig. 2B). In addition, ***R*-Eu-Et-1** in the aggregated state is connected by five different strong hydrogen bonds C−H•••O bonds to form a three-dimensional structure (Fig. 2C). First, hydrogen bond I ($C_{50}−H_{50C}$•••$O_{13}$, 2.553 Å) connects multiple independent units to form a chain structure with an AABB arrangement. Second, chains are connected by hydrogen bonds II ($C_{20}−H_{20}$•••$O_{15}$, 2.493 Å), III ($C_{46}−H_{46}$•••$O_6$, 2.504 Å), and IV ($C_{13}−H_{13}$•••$O_{15}$, 2.554 Å) to become a layered structure. Finally, chains and chains and layers are connected by hydrogen bonds V ($C_{43}−H_{43}$•••$O_6$, 2.484 Å) to form aggregated ***R*-Eu-Et-1**. Note that a strong hydrogen bond (II-V) exists between the hydrogen atom on the molecular rotor and the hydrogen bond acceptor oxygen atom on the nitrate, and this bond fixes the molecular conformation of ***R*-Eu-Et-1** and locks the molecular rotor. The introduction of a molecular rotor structure can help facilitate the construction of artificial intelligence molecules, and as far as we know, artificial intelligence molecules based on lanthanide complexes are very rare. Structural analysis shows that the $Eu^{III}$ ion in the ***R*-Eu-Et-1** structure is in the $O_6N_4$ coordination environment jointly provided by *R*-L[1] and $NO_3^-$. *SHAPE* calculates that the coordination configuration of the above $Eu^{III}$ ion is sphenocorona with a $C_{2v}$ symmetric environment (Supplementary Fig. 1D and Supplementary Table 2). In the ***R*-Eu-Et-1** structure, the coordination mode of the ligand *R*-L[1] is as follows: $\mu_1$-$\eta^1$:$\eta^1$:$\eta^1$:$\eta^1$ (Supplementary Fig. 1B). Topological analysis shows that ***R*-Eu-Et-1** can serve as an eight-connected *fcu* network with a distance ranging from 10.701 to 14.482 Å (Fig. 2D). All Ln−O/N bond lengths are within the normal range (Supplementary Data 1).

A pair of enantiomers ***R/S*-Eu-Et-2** was obtained under the same reaction conditions by replacing $(1R/S,2R/S)$-(-/+)-cyclohexanediamine with an equimolar ratio of $(1R/S,2R/S)$-(-/ + )−1,2-

diphenylethylenediamine (Supplementary Fig. 1A and Data 1). SCXRD indicates that **R/S-Eu-Et-2** crystallizes in the orthorhombic space group $P2_12_12_1$ (Supplementary Table 1). As shown in Fig. 2E, R/**S-Eu-Et-2** consists of an $Eu^{III}$ ion, a chiral Schiff base ligand (1E,1′E)-N,N′-(cyclo-hexane-1,2-diyl)bis(1-(1-ethyl-1H-imidazol-2-yl)methanimine) $(R-L^2)$ with multidentate chelating coordination, three terminally coordinated $NO_3^-$ ions, and a free $CH_3CN$ molecule, and its molecular formula is $[Eu(R-L^2)(NO_3)_3]CH_3CN$. Interestingly, in the molecular state of **R-Eu-Et-2**, the cyclohexane on $R-L^2$ exhibits intense vibration because of the configuration change. However, in the aggregated state, the config-uration switching of cyclohexane in **R-Eu-Et-2** is hindered by the $-CH_2CH_3$ of $R-L^2$ and the oxygen atoms of $NO_3^-$ on the adjacent molecule, making it difficult to effectively vibrate (Fig. 2F). **R-Eu-Et-2** in the aggregated state consists of four kinds of C−H•••O strong hydrogen bonds to form a three-dimensional structure (Fig. 2G). First, it forms a chain structure by linking multiple independent units through hydrogen bond VI ($C_8-H_{8B}$ •••$O_{36}$, 2.845 Å). Second, multiple chains are connected by hydrogen bond VII ($C_{17}-H_{17}$ •••$O_4$, 3.046 Å) to become a layered structure. At last, the aggregated **R-Eu-Et-2** is formed

by linking individual chains and layers through VIII ($C_{11}-H_{11A}$ •••$O_{33}$, 2.811 Å) and IX ($C_9-H_{9B}$ •••$O_4$, 3.043 Å). In addition, the hydrogen atom on the $R-L^2$ ligand forms a C−H•••N hydrogen bond (X, 3.945 Å) with the hydrogen bond acceptor nitrogen atom on the $CH_3CN$. Notably, the hydrogen atoms on the cyclohexane vibration unit have strong hydrogen bond interactions with the hydrogen bond acceptor oxygen atoms on the nitrate (VI, VIII, and IX), a feature which helps to fix the **R-Eu-Et-2** molecular conformation and lock the molecular vibration. To the best of our knowledge, this work constructs dynamic rare earth complexes based on molecular vibrations. Structural ana-lysis shows that the $Eu^{III}$ ion in the **R-Eu-Et-2** structure is in the $O_6N_4$ coordination environment jointly provided by $R-L^2$ and $NO_3^-$. The coordination configuration of the $Eu^{III}$ ions and the coordination mode of the ligand ($R-L^2$) of **R-Eu-Et-2** are consistent with those of **R-Eu-Et-1** (Supplementary Fig. 1E and Table 3). In the **R-Eu-Et-2** structure, the coordination mode of the ligand $R-L^2$ is as follows: $\mu_1$-$\eta^1$:$\eta^1$:$\eta^1$:$\eta^1$ (Sup-plementary Fig. 1C).Topological analysis shows that **R-Eu-Et-2** can serve as a six-connected *fcu* network with a distance ranging from 10.480 to 14.286 Å (Fig. 2H).

## A. Restriction of Intramolecular Rotation

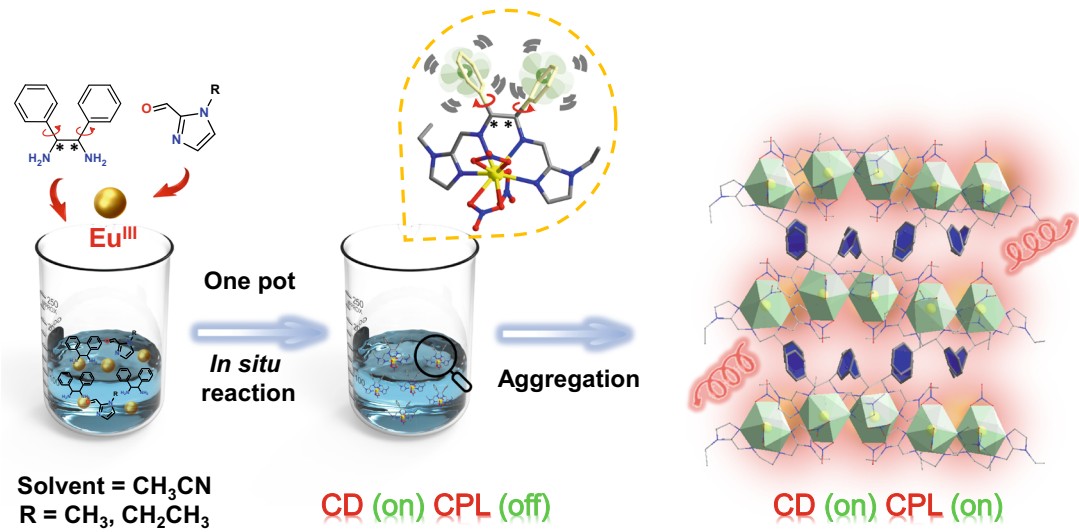

## B. Restriction of Intramolecular Vibration

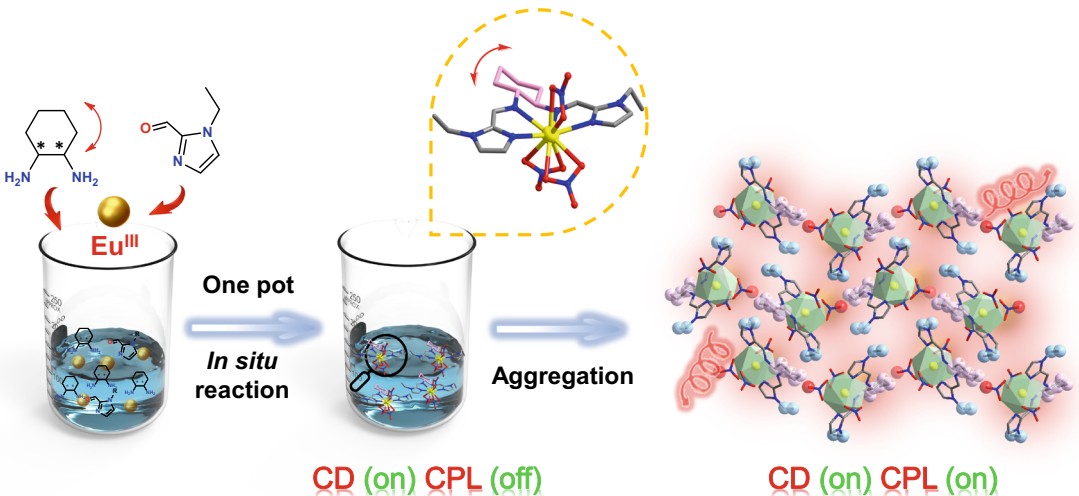

**Fig. 1 | Synthesis of dynamic lanthanide complexes.** Schematic diagram of the construction of dynamic lanthanide complexes with RIR (**A**) and RIV (**B**) properties under one-pot conditions using ligands with molecular rotor structures and vibrational units. The golden balls represent $Eu^{III}$ ions; The red shadow in the pic-ture is the red light emitted by the complex.

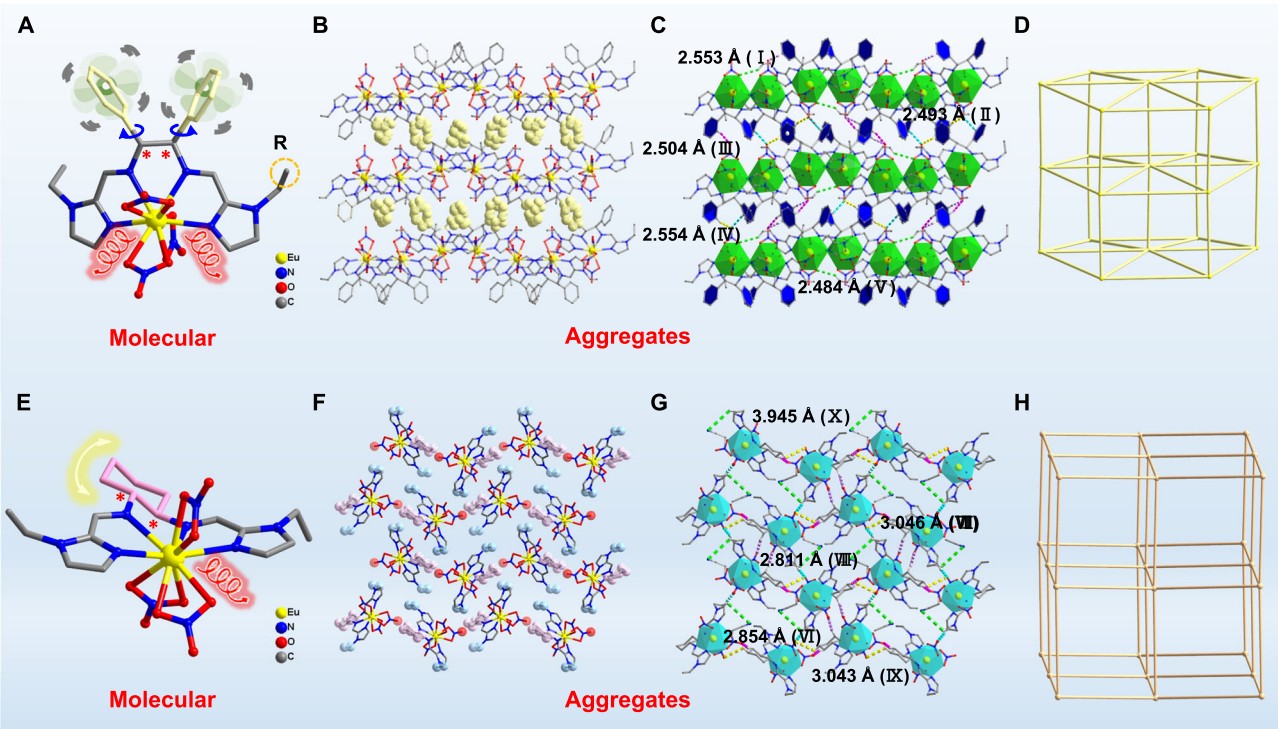

**Fig. 2 | Structural analysis. A, E** Molecular structures of ***R*-Eu-R-1** (R = Et/Me) and ***R*-Eu-Et-2**. **B, F** Schematic diagram of the free rotation/vibration restriction of the benzene ring/cyclohexane of ***R*-Eu-Et-1/2** in the aggregated state. **C, G** Hydrogen bonding of ***R*-Eu-Et-1/2**. **D, H** Topological structure of ***R*-Eu-Et-1/2**. Note: The blue and yellow arrows represent the rotation/vibration of the molecular rotor.

(1*R*,2*R*)-(-)-1,2-diphenylethylenediamine was changed to ethylenediamine, and under the same synthesis conditions, a Eu$^{III}$ complex **Eu-Et-3** was successfully obtained which does not have a rotor and vibrational primitives and has the same coordination environment as ***R*-Et-Eu-1** (Supplementary Fig. 2). SCXRD results show that **Eu-Et-3** crystallizes in the $P\bar{1}$ space group of the triclinic system (Supplementary Table 1). As shown in Supplementary Fig. 2A, **Eu-Et-3** consists of a Eu$^{III}$ ion and a Schiff base ligand L$^3$ ((1E,1'E)-N,N'-(ethane-1,2-diyl)bis(1-(1-ethyl-1H-imidazol-2-yl)methanimine)) and three end-group coordinated NO$_3^-$ ions, its molecular formula is [Eu(L$^3$)(NO$_3$)$_3$]. **Eu-Et-3** is connected by six kinds of C–H•••O hydrogen bonds to form a 3D stacking structure (Supplementary Fig. 2B, C). First, hydrogen bonds I (C$_{19}$–H$_{19}$•••O$_8$, 2.484 Å) and II (C$_{32}$–H$_{32B}$•••O$_{12}$, 2.733 Å) connect multiple independent units to form a chain structure. Secondly, the chains are connected through hydrogen bonds III (C$_{21}$–H$_{21B}$•••O$_4$, 2.860 Å) and IV (C$_{31}$–H$_{31C}$•••O$_{13}$, 2.869 Å) to form a layered structure. Finally, the aggregated **Eu-Et-3** is formed by connecting chains to chains and layers to layers through hydrogen bonds V (C$_{33}$–H$_{33B}$•••O$_4$, 2.837 Å) and VI (C$_{30}$–H$_{30A}$•••O$_{13}$, 2.734 Å). Structural analysis shows that the Eu$^{III}$ ion in the **Eu-Et-3** structure is in the O$_6$N$_4$ coordination environment provided by L$^3$ and NO$_3^-$ ions. *SHAPE* calculated that the coordination configuration of the above Eu$^{III}$ ion is tetradecahedron with $C_{2v}$ symmetry environment (Supplementary Fig. 2E and Table 4). In the **Eu-Et-3** structure, the coordination mode of ligand L$^3$ is: $\mu_1$-$\eta^1$:$\eta^1$:$\eta^1$:$\eta^1$ (Supplementary Fig. 2D). Topological analysis shows that **Eu-Et-3** can be regarded as an eight-connected *fcu* network with a distance range of 7.859 to 10.844 Å (Supplementary Fig. 2F).

***R*/*S*-Eu-R-1** (R = Et/Me) and ***R*/*S*-Eu-Et-2**, which are enantiomers, have same Fourier transform infrared (FTIR) characteristic absorption peaks at the similar position (Supplementary Fig. 3). Thermal stability tests of ***R*/*S*-Eu-R-1** (R = Et/Me) and ***R*/*S*-Eu-Et-2** were carried out under a flowing nitrogen atmosphere at a rate of 5 °C min$^{-1}$ slowly ramped up from 35 °C to 1000 °C (Supplementary Fig. 4). All the experimental results show that ***R*/*S*-Eu-R-1** (R = Et/Me) and ***R*/*S*-Eu-Et-2** have excellent

thermal stability. The SEM images of ***R*/*S*-Eu-Me-1**, ***R*/*S*-Eu-Et-1**, and ***R*/*S*-Eu-Et-2** clearly show that they are all bulk crystals, and the surfaces are very clean (Supplementary Fig. 5). In addition, ***R*/*S*-Eu-R-1** and ***R*/*S*-Eu-Et-2** have similar polycrystalline powder X-ray diffraction (PXRD) spectra, indicating that they are pure phases (Supplementary Fig. 6).

## Aggregation-induced Emission of *R*/*S*-Eu-R-1 (R = Et/Me) and *R*/*S*-Eu-Et-2

A variety of AIEgens, AIEgens-based MOFs, and clusters with AIE behavior have been designed and synthesized in recent years, and they have shown broad application prospects in the fields of sensing, multiple anti-counterfeiting, solid-state lighting, and biological diagnosis and treatment[41-44]. At present, molecules with AIE properties are mainly designed and constructed based on restricted RIM, and the common strategy involves introducing structural units that can rotate or vibrate freely. According to the structural characteristics of ***R*-Eu-Et-1/2**, further studies were conducted on their AIE properties. Specifically, by using glycerin as a poor solvent, the same mass of ***R*/*S*-Eu-Et-1** and ***R*/*S*-Eu-Et-2** were dissolved in glycerin/DMSO mixed solutions with different glycerin contents ($f_w$), and their emission spectra were tested. The experimental results showed that with the increase of DMSO, the emission intensity of ***R*/*S*-Eu-Et-1** and ***R*/*S*-Eu-Et-2** gradually weakened. Moreover, with the increase of the content of glycerin, the emission intensity of the solution increases gradually, and strong luminescence was observed when the content of glycerin was 99%, indicating that both ***R*/*S*-Eu-Et-1** and ***R*/*S*-Eu-Et-2** have typical AIE characteristics (Fig. 3). Calculated by $I/I_0$, the $\alpha_{AIE}$ values of ***R*/*S*-Eu-Et-1** and ***R*/*S*-Eu-Et-2** are 92.54/87.95 and 13.44/16.8 (Fig. 3). To further verify the AIE behavior of ***R*/*S*-Eu-Et-1**, ***R*/*S*-Eu-Et-2**, and ***R*/*S*-Eu-Me-1**, we tested the QYs of ***R*/*S*-Eu-Et-1** and ***R*/*S*-Eu-Et-2**, and ***R*/*S*-Eu-Me-1** in glycerin/DMSO or CH$_3$CN/DMF mixed solutions with different ratios. Experimental results show that as the proportion of poor solvent glycerin or CH$_3$CN continues to increase, the luminescence QY of these Eu$^{III}$ complexes continues to increase. When the glycerin or CH$_3$CN content is 99%, the

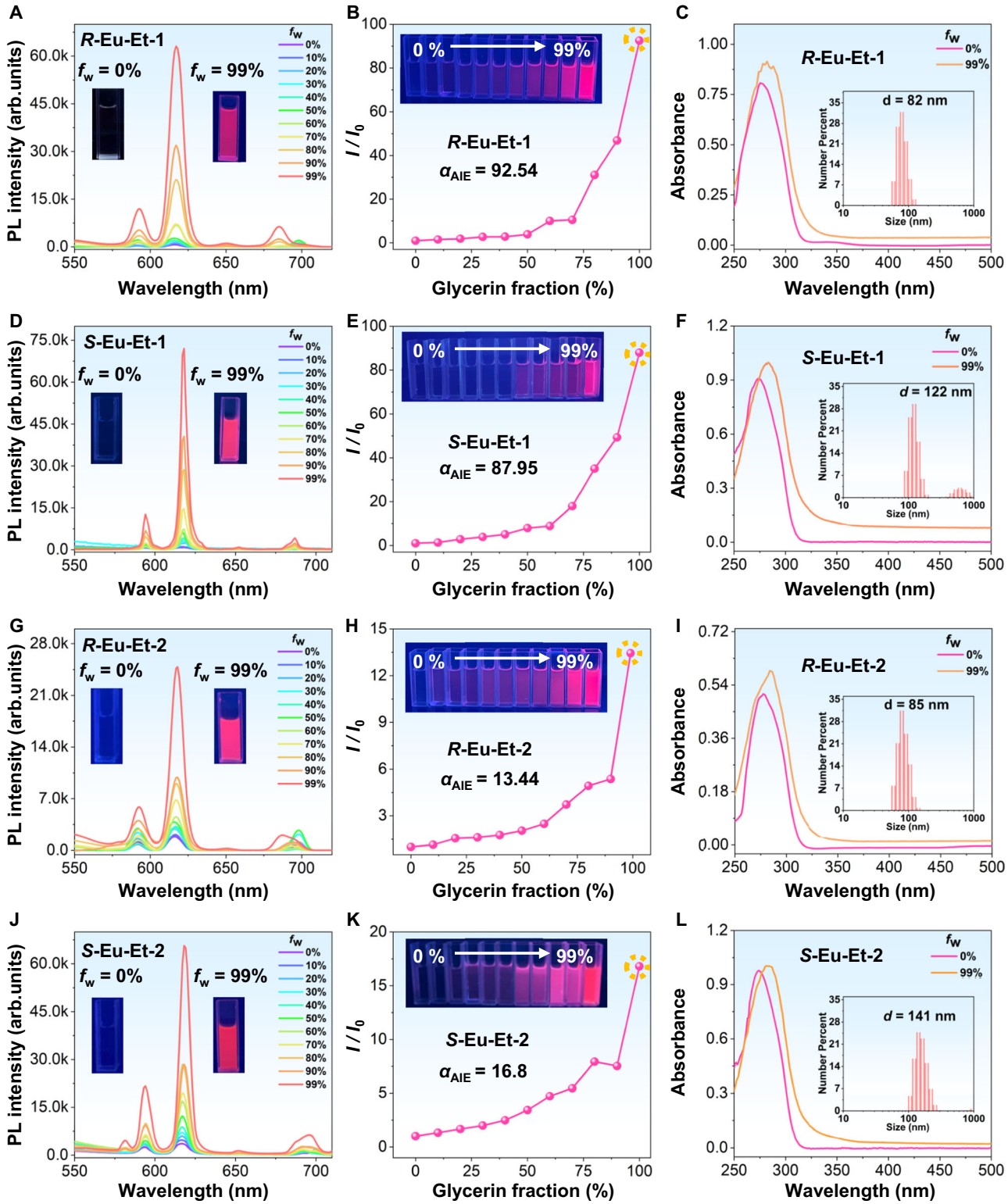

**Fig. 3 | Characterization of AIE properties. A, D, G, J** Emission spectra of **R/S-Eu-Et-1** and **R/S-Eu-Et-2** in different contents of glycerin/DMSO mixed solutions under excitation at 365 nm, the inset shows the luminescence photos of **R/S-Eu-Et-1** and **R/S-Eu-Et-2** in 0 and 99% glycerin/DMSO mixed solutions respectively under UV-light. **B, E, H, K** Luminescence intensities of **R/S-Eu-Et-1** and **R/S-Eu-Et-2** at 617 nm as a function of $f_w$, the inset shows the photoluminescence of **R/S-Eu-Et-1** and **R/S-Eu-Et-2** in different proportions of glycerin/DMSO mixed solutions under UV-light.

**C, F, I, L** Absorption spectra of **R/S-Eu-Et-1** and **R/S-Eu-Et-2** in glycerin/DMSO mixed solutions with different contents of $f_w$, the insets show the DLS results of **R/S-Eu-Et-1** and **R/S-Eu-Et-2** in 99% glycerin/DMSO mixture. Note: The orange dotted circle represents the luminous intensity position of **R/S-Eu-Et-1** and **R/S-Eu-Et-2** under the condition of mixed solution (99% glycerin: 1% DMSO): (0% glycerin: 100% DMSO); $f_w$ represents the ratio of glycerin to DMSO.

QYs of $R/S$-Eu-Et-1 and $R/S$-Eu-Et-2, and $R/S$-Eu-Me-1 reach the maximum, which are 27.2/27.9, 27.8/24.1, 25.0/28.0, 26.3/26.7, 18.6/19.9, and 29.5/29.9, respectively, once again proving that they have significant aggregation promotes antenna effect behavior (Supplementary Fig. 7−9 and Data 2). In addition, only the metal center $Eu^{III}$ ion was replaced by $Tb^{III}/Gd^{III}$ ion and the isostructural $R/S$-Gd-Et-1, $R/S$-Gd-Me-1, $R$-Gd-Et-2, $R$-Tb-Et-1, $R/S$-Tb-Me-1, $R$-Tb-Et-2 were obtained respectively under the same conditions as $R$-Eu-Et-1 (Supplementary Table 1 and Fig. 10). Since the energy levels of $R$-$L^1$ and $R$-$L^2$ do not match the energy levels of $Tb^{III}/Gd^{III}$ ions and the antenna effect cannot be performed, $R/S$-Gd-Et-1, $R/S$-Tb-Et-1, $R/S$-Gd-Et-2, $R/S$-Tb-Et-2, $R$-Gd-Me-1, $R$-Tb-Me-1 all exhibit fluorescence of the ligands. The emission spectra of the above lanthanide complexes were tested in mixed solutions of glycerin/DMSO with different ratios ($f_w$). The results show that $R/S$-Gd-Et-1, $R/S$-Tb-Et-1, $R/S$-Gd-Et-2, $R/S$-Tb-Et-2, $R$-Gd-Me-1, $R$-Tb-Me-1 and all exhibit obvious AIE behaviors (Supplementary Fig. 11−13). Meanwhile, we investigated the AIE properties of $R/S$-Eu-Me-1, and $S$-Eu-Et-1/2 in $CH_3CN$/DMF mixtures by using $CH_3CN$ as a poor solvent. The experimental results show that the emission intensity of $R/S$-Eu-Me-1, and $S$-Eu-Et-1/2 is very weak in pure DMF solution (Supplementary Fig. 14 and 15). In the mixed solution with glycerin content of 99%, the UV-Vis absorption spectra of $R/S$-Eu-Et-1 and $R/S$-Eu-Et-2 obviously do not start from the origin but have a certain absorption value (Fig. 3C, F, I, L), which shows that they have formed an aggregated suspension in glycerin, and the above absorption value is given by aggregate scattering results. In addition, the UV-Vis absorption of $R/S$-Eu-Me-1 and $S$-Eu-Et-1/2 has a certain degree of red shift, which can be attributed to the size change of the complexes in the aggregated and dispersed states[24,45–47]. $R/S$-Gd-Et-1, $R/S$-Gd-Et-2, $R/S$-Tb-Et-1, $S$-Eu-Et-1/2, and $R/S$-Eu-Me-1 have the same result (Supplementary Fig. 11, 12, 14, 15). The particle sizes of the aggregate's $R/S$-Eu-Et-1 and $R/S$-Eu-Et-2 in glycerin were measured by dynamic light scattering as 82/122 and 85/141 nm, respectively (Fig. 3C, F, I, L illustrations). In addition, the particle sizes of the aggregate's $R/S$-Eu-Me-1 and $S$-Eu-Et-1/2 in $CH_3CN$ were measured by dynamic light scattering as 712/615, and 390/396 nm, respectively (Supplementary Fig. 14, 15 illustrations). In addition, the Zeta potentials of $R/S$-Eu-Et-1, $R/S$-Eu-Et-2, and $R/S$-Eu-Me-1 in the aggregated state are -21.4/-24.23, -16.93/-20.13, and -21.63/-21.06 mV, respectively, indicating that their aggregates can remain stable (Supplementary Fig. 16). Using $CH_3CN$ as a poor solvent, Eu-Et-3 was dissolved in $CH_3CN$/DMF mixed solutions with different $CH_3CN$ contents ($f_w$), and the emission spectra of the above solutions were tested. Experimental results show that as the proportion of $CH_3CN$ increases, the emission intensity of Eu-Et-3 rapidly weakens, indicating that Eu-Et-3 has an ACQ effect (Supplementary Fig. 17). Furthermore, when we excited $R/S$-Eu-R-1 (R = Et/Me) and $R/S$-Eu-Et-2 dispersed in $CH_3CN$ solution with light at 365 nm, they both showed 592, 617, 650, and 685 nm, which are assigned to the $^5D_0 \rightarrow ^7F_1$, $^5D_0 \rightarrow ^7F_2$, $^5D_0 \rightarrow ^7F_3$, and $^5D_0 \rightarrow ^7F_4$ energy level transitions of $Eu^{III}$ ions, respectively (Supplementary Fig. 18). The luminescence lifetimes of $R/S$-Eu-R-1 (R = Et/Me) and $R/S$-Eu-Et-2 dispersed in $CH_3CN$ are as long as $\tau_{1/e}$ = 1207.5 (± 0.022), 1225.9 (± 0.027), 1223.6 (± 0.031), 1246.4 (± 0.028), 1240.8 (± 0.047), and 1237.5 (± 0.023) μs (Supplementary Fig. 19). The QYs of $R/S$-Eu-R-1 and $R/S$-Eu-Et-2 dispersed in $CH_3CN$ are 26%/26% (R = Et), 29%/28% (R = Me), and 17%/17%, respectively (Supplementary Fig. 20).

The AIE properties of $R$-Eu-Et-1 can be explained by the RIR mechanism. In the single-molecule state, the ligand $R$-$L^1$ in the structure $R$-Eu-Et-1 has two molecular rotors that can rotate freely, so the excited state molecules mainly return to the ground state through non-radiative transitions, which leads to the shutdown of the ISC process and the ineffective ET process. Thus, the radiative transition channel of the lanthanide metal ion is closed. In the aggregated state, the rotation of the $R$-Eu-Et-1 molecular rotor is greatly restricted by the steric hindrance of adjacent molecules and strong hydrogen bonding,

thereby closing the non-radiative transition channel of the excited state, opening the ISC process, and undertaking the ET process (antenna effect) to facilitate the radiative transition of lanthanide metal ions, consequently showing the characteristic emission of $Eu^{III}$ ions. This work marks that aggregation-enhanced antenna effects promoted characteristic radiative transitions in lanthanide emitters.

As no obvious molecular rotor structure exists in the $R$-Eu-Et-2 structure, its AIE properties may be derived from the rare RIV mechanism. In the unimolecular state, the cyclohexane moiety on the ligand $R$-$L^2$ in the $R$-Eu-Et-2 structure exhibits rapid free vibrations because of configuration changes, and the excited state energy returns to the ground state mainly through non-radiative transitions. In the aggregated state, the strong hydrogen bonding leads to the obvious restriction of the free vibration of the cyclohexane part in the $R$-Eu-Et-2 structure, and the non-radiative transition channel of the excited state energy is closed, thereby promoting the energy transfer process based on the antenna effect and opening the characteristic emission of $Eu^{III}$ ions. Thus far, lanthanide complex emitters are usually constructed by lanthanide metal ions and organic ligands. When a specific metal ion is selected, its luminescence mainly depends on the organic ligand with a specific connection and configuration[14]. Therefore, the luminescence performance of lanthanide complex emitters primarily depends on the complex ligand structure design, and achieving dynamic luminescence performance is challenging. Introducing molecular rotor structures or high-frequency vibration units into lanthanide complex emitters is an effective way to construct dynamic luminescent complexes with lanthanide emission.

## Solid state luminescence and circularly polarized luminescence of $R/S$-Eu-R-1 (R = Et/Me) and $R/S$-Eu-Et-2

The complex $4f^n$ electron shell structure endows the lanthanide complexes with extraordinaryoptical properties[1–4]. The bright red luminescence of $R$-Eu-Et-1/2 solids was clearly observed under 365 nm UV-light conditions (Fig. 4A). In addition, the bright red emission of $R$-Eu-Et-1 was captured using a confocal laser scanning microscope (CLSM) under the excitation condition of 405 nm (Supplementary Fig. 21). As shown in Supplementary Fig. 22, the CIE coordinates of $R/S$-Eu-R-1 (R = Et/Me) and $R/S$-Eu-Et-2 are (0.6705, 0.3292), (0.6707, 0.3290), (0.6701, 0.3297), (0.6674, 0.3323), (0.6676, 0.3322), and (0.6671, 0.3326), respectively, all in within the red-light range. To deeply explore the luminescent behavior of the dynamic chiral luminescent complex's $R/S$-Eu-R-1 and $R/S$-Eu-Et-2 in the aggregated state, their solid-state luminescence spectra were tested. When $R/S$-Eu-R-1 (R = Et/Me) and $R/S$-Eu-Et-2 were excited by 365 nm, they all showed characteristic emission peaks at 592, 617, 650, and 685 nm, which are respectively attributed to the characteristic energy level transitions of $^5D_0 \rightarrow ^7F_1$, $^5D_0 \rightarrow ^7F_2$, $^5D_0 \rightarrow ^7F_3$, and $^5D_0 \rightarrow ^7F_4$ of $Eu^{III}$ ions (Fig. 4A and Supplementary Fig. 23). Similarly, when Eu-Et-3 is excited at 365 nm, it exhibits characteristic emission peaks at 593, 617, 650, and 687 nm, which can be attributed to the characteristic energy level transition of $^5D_0 \rightarrow ^7F_{1-4}$ of $Eu^{III}$ ions (Supplementary Fig. 24). In addition, the solid-state luminescent QYs of $R/S$-Eu-R-1 (R = Et/Me), $R/S$-Eu-Et-2, and Eu-Et-3 were tested, and they were as high as 41%/41% (R = Et), 42%/42% (R = Me), 36%/35%, and 4.2%, respectively (Supplementary Fig. 25). It is worth noting that the solid-state luminescence QYs of $R$-Eu-Et-1/2 with rotors or vibration units are much higher than that of Eu-Et-3, where the QY of $R$-Eu-Et-1/2 are about 10 and 9 times that of Eu-Et-3, respectively. Interestingly, $R$-Eu-Et-1/2 were still observed to have instantaneous bright solid-state luminescence when the 365 nm UV-light was turned off (Supplementary Fig. 26 and Supplementary Movie 1–4), indicating their long luminescence lifetimes. Then, the solid-state luminescence lifetimes of $R/S$-Eu-R-1 (R = Et/Me) and $R/S$-Eu-Et-2 were tested, and the lifetimes were as long as $\tau_{1/e}$ = 934.4 (± 0.009)/929.5 (± 0.008), 979.3 (± 0.009)/977.0 (± 0.008) and 1112.9 (± 0.021)/1039.8 (± 0.009) μs, respectively (Fig. 4B and

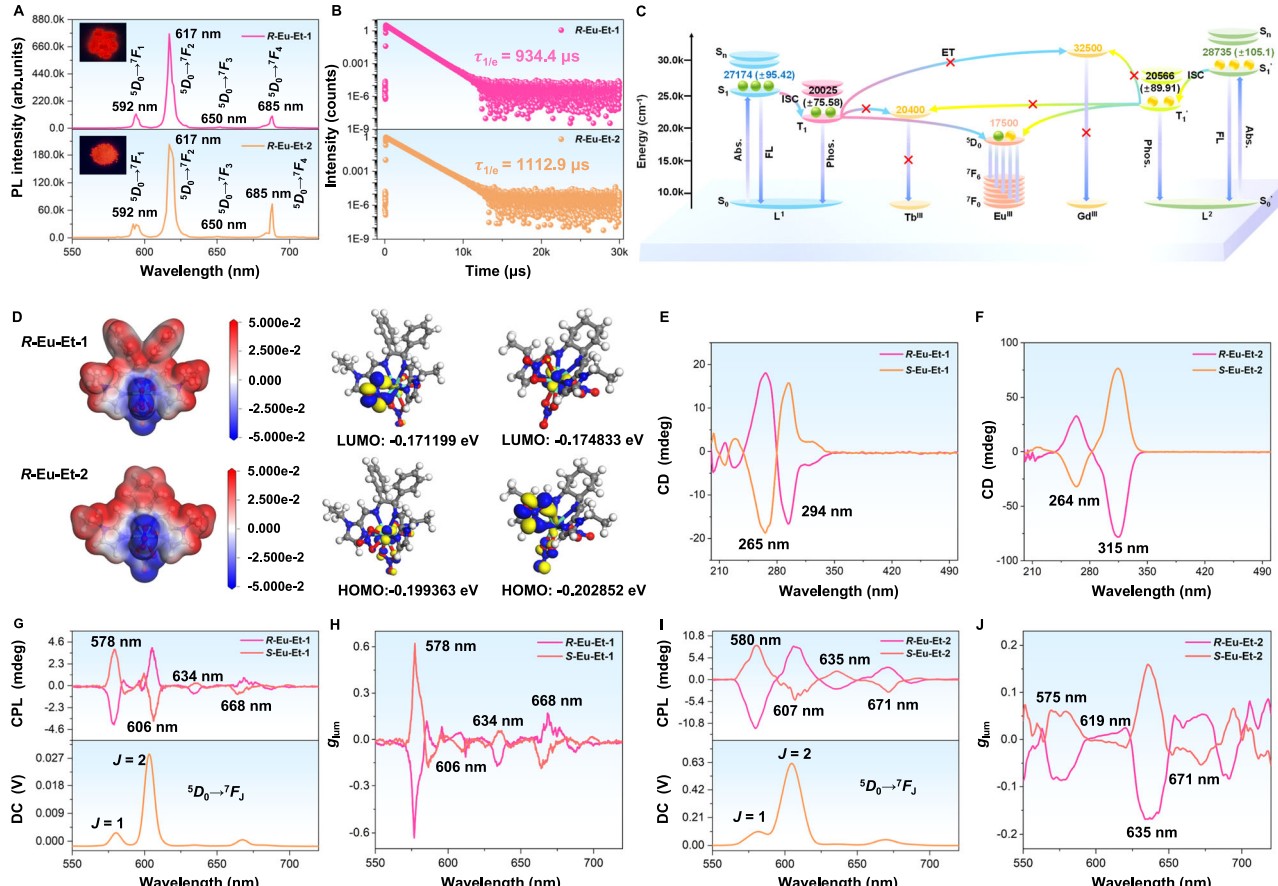

**Fig. 4 | Photophysical properties and theoretical calculations. A** Solid-state luminescence spectrum under excitation at 365 nm (*D* represents the quintet; *F* represents the septet). **B** The decay curve of $^5D_0$ energy level in solid-state *R*-**Eu**-**Et**-**1/2**. **C** Schematic diagram of the Jablonski energy levels in different states. **D** The electron cloud density maps of *R*-**Eu**-**Et**-**1/2** by DFT theoretical calculation (GGA-PBE/DND). **E**, **F** CD spectra of *R/S*-**Eu**-**Et**-**1** and *R/S*-**Eu**-**Et**-**2** dispersed in $CH_3CN$ solution. **G**, **I** CPL and DC spectra. **H**, **J** corresponding $g_{lum}$ values. Note: The green and yellow balls represent excitons; Abs. Absorption, FL fluorescence, Phos. Phosphorescence, ISC intersystem crossing, ET energy transfer.

Supplementary Fig. 27). In addition, the lifetime of **Eu**-**Et**-**3** is as long as $\tau_{1/e} = 1022.8$ ($\pm 0.032$) μs (Supplementary Fig. 28). Therefore, the above series of luminescent results reveal that the chiral luminescent $Eu^{III}$ complexes *R/S*-**Eu**-**R**-**1** and *R/S*-**Eu**-**Et**-**2** are both excellent solid-state emitters.

In order to deeply explore the energy transfer pathway of *R*-**Eu**-**Et**-**1/2**, the Jablonski energy level diagram was further drawn (Fig. 4C). According to Reinhoudt's rule of thumb[21,48,49], when the energy level difference between the excited singlet state and the excited triplet state of the ligand is greater than 5000 cm$^{-1}$, the intersystem crossing (ISC) process can be guaranteed to be effected. At room temperature, (1*R/S*,2*R/S*)-(-/+)−1,2-diphenylethylenediamine and (1*R/S*,2*R/S*)-(-/+)-cyclohexanediamine were stirred with 1-Ethyl-2-imidazolecarboxaldehyde respectively, in 2 mL of $CH_3OH$ solution for 24 h to obtain *R/S*-**L**$^1$ and *R/S*-**L**$^2$ (Supplementary Fig. 29 and Table 5). According to the solid-state UV-Vis absorption spectra of *R*-**L**$^1$ and *R*-**L**$^2$, it can be seen that the energy values of the excited singlet states $S_1$ and $S_1'$ of *R*-**L**$^1$ and *R*-**L**$^2$ are 27174 ($\pm 95.42$) and 28735 ($\pm 105.1$) cm$^{-1}$ respectively (Supplementary Fig. 30A, B). Obviously, the lowest excited state energy level of $Gd^{III}$ ions (32500 cm$^{-1}$) is significantly higher than the excited singlet energy levels of *R*-**L**$^1$ and *R*-**L**$^2$, and energy cannot be transferred from the ligand to $Gd^{III}$ ions. Therefore, the phosphorescence spectrum of *R*-**Gd**-**Et**-**1/2** was collected at 77 K, and the energy values of the excited triplet states $T_1$ and $T_1'$ of *R*-**L**$^1$ and *R*-**L**$^2$ were obtained by Gaussian curve fitting to be 20025 ($\pm 75.58$) and 20566 ($\pm 89.91$) cm$^{-1}$, respectively (Supplementary Fig. 30C, D). The $\Delta E_L$ between the excited singlet state and

excited triplet state of *R*-**L**$^1$ and *R*-**L**$^2$ was calculated to be 7149 ($\pm 19.84$) and 8169 ($\pm 15.19$) cm$^{-1}$ respectively. Obviously, the above $\Delta E_L$ are all higher than 5000 cm$^{-1}$, indicating that *R*-**L**$^1$ and *R*-**L**$^2$ can effectively carry out the ISC process. The energy gap ($\Delta E$) between the excited triplet energy level of the ligand and the lowest excited state energy level of $Ln^{III}$ must be in a suitable range for effective energy transfer process[50,51]. According to Latva's rule of thumb, a $\Delta E$ of 2000–5000 cm$^{-1}$ is optimal for the energy transfer process of $Eu^{III}$, while the optimal $\Delta E$ of energy transfer for $Tb^{III}$ ions is 2400 ± 300 cm$^{-1}$ [52–54]. In addition, the energy value of the $^5D_0$ energy level of $Eu^{III}$ ions is 17500 cm$^{-1}$, and the energy value of the $^5D_4$ energy level of $Tb^{III}$ is 20400 cm$^{-1}$. Further calculations show that the $\Delta E$ between the excited triplet states of *R*-**L**$^1$ and *R*-**L**$^2$ and the $^5D_0$ of $Eu^{III}$ ions are 2525 ($\pm 75.58$) and 3066 ($\pm 89.91$) cm$^{-1}$ respectively. Obviously, both *R*-**L**$^1$ and *R*-**L**$^2$ can efficiently sensitize the luminescence of $Eu^{III}$ ions. In addition, the $\Delta E$ between the excited triplet states of *R*-**L**$^1$ and *R*-**L**$^2$ and the $^5D_4$ of $Tb^{III}$ ions are −375 ($\pm 75.58$) and 166 ($\pm 89.91$) cm$^{-1}$ respectively, indicating that they are unable to transfer energy to $Tb^{III}$ ions through effective antenna effects (Fig. 4C). Density functional theory calculations (DFT; functional/basis set are GGA-PBE/DND) were used to obtain the electron cloud distribution and the lowest unoccupied molecular orbital (LUMO) and highest occupied molecular orbital (HOMO) (Fig. 4D and Supplementary Data 3 and 4). Clearly, the main concentration of electron clouds on organic ligands demonstrates their great potential as antennas[55]. In addition, the electron cloud on the organic ligand obviously shifts to the metal center $Eu^{III}$ ions under excited state

conditions. The absorption of polarized light by two pairs of enantiomers **R/S-Eu-Et-1** and **R/S-Eu-Et-2** in the aggregated state was measured using circular dichroism (CD) absorption spectroscopy. As shown in Fig. 4E, F, both **R/S-Eu-Et-1** and **R/S-Eu-Et-2** exhibit good mirror image CD curves in the aggregated state, indicating that the **R/S-Eu-Et-1** and **R/S-Eu-Et-2** are all enantiomers of each other. Furthermore, their Cotton effects at 265/264 nm (**R/S-Eu-Et-1**) and 294/315 nm (**R/S-Eu-Et-2**) may be attributed to the characteristic absorption of the diphenyl unit. Therefore, **R/S-Eu-Et-1** and **R/S-Eu-Et-2** in the aggregate state can show specific chiralities.

CD and luminescence are two key factors in the construction of CPL materials[56]. In addition, two key indicators to characterize the performance of CPL are $g_{lum}$ and $B_{CPL}$ ($g_{lum} = 2(I_L - I_R)/(I_L + I_R)$, where $I_L$ and $I_R$ are the intensities of emitted left and right circularly polarized light, respectively). The CPL brightness is given by $B_{CPL} = \varepsilon_\lambda \cdot \Phi \cdot |g_{lum}|/2$, where $\varepsilon_\lambda$ is the molar absorptivity at the excitation wavelength and $\Phi$ is the QY[31]. However, high $B_{CPL}$ values can be obtained from strongly absorbing molecules with relatively low asymmetry factors, whereas unusually strong asymmetry factors are hampered by weak molar absorptivity and QY. Therefore, taking both $g_{lum}$ and $B_{CPL}$ into consideration and realizing their double promotion is challenging[31]. Lanthanide metal ions typically exhibit high $g_{lum}$ due to possible magnetic dipole-allowed transitions. In addition, the characteristic radiative transition of lanthanide emitters can be promoted by enhancing the antenna effect to obtain a high QY $\Phi$. Therefore, lanthanide complexes are one of the best candidates for developing CPL materials. In particular, the Eu$^{III}$ complexes, whose $^5D_0 \to {}^7F_1$ energy level has high $g_{lum}$, have attracted extensive attention. Based on the CD and PL properties of **R/S-Eu-R-1** (R = Et/Me) and **R/S-Eu-Et-2**, we further tested the CPL performance of the above complexes. Both **R/S-Eu-R-1** (R = Et/Me) and **R/S-Eu-Et-2** dispersed in CH$_3$CN solution exhibited distinct mirror image CPL signals (Fig. 4G, I, and Supplementary Fig. 31). Specifically, obvious mirror image CPL signal peaks were observed at 578, 606, 634, and 668 nm for **R/S-Eu-Et-1**, which were assigned to the $^5D_0 \to {}^7F_1$, $^5D_0 \to {}^7F_2$, $^5D_0 \to {}^7F_3$, and $^5D_0 \to {}^7F_4$ energy level transitions of Eu$^{III}$ ions, respectively (Fig. 4G). According to $g_{lum} = 2(I_L-I_R)/(I_L+I_R)$, the corresponding $g_{lum}$ values of **R/S-Eu-Et-1** at 578, 606, 634, and 668 nm were calculated as 0.64/0.62, 0.06/0.08, 0.17/0.05 and 0.17/0.18, respectively (Fig. 4H). According to $B_{CPL} = \varepsilon_\lambda \cdot \Phi \cdot |g_{lum}|/2$, the corresponding $B_{CPL}$ values of **R/S-Eu-Et-1** at 578, 606, 634, and 668 nm were calculated as 2429/2353, 228/304, 645/190 and 645/683 M$^{-1}$cm$^{-1}$, respectively (Supplementary Table 6 and the absorbance of the corresponding molar absorptivity $\varepsilon_\lambda$ is obtained from Supplementary Fig. 32). Similarly, **R/S-Eu-Me-1** exhibited obvious image CPL signal peaks at the same wavelength, which were attributed to the $^5D_0 \to {}^7F_{1-4}$ energy level transition of Eu$^{III}$ ions (Supplementary Fig. 31A). The corresponding $g_{lum}$ values of **R/S-Eu-Me-1** at 578, 606, 635, and 668 nm are 0.62/0.49, 0.08/0.14, 0.14/0.16, and 0.18/0.13, respectively (Supplementary Fig. 31B). The $B_{CPL}$ values were 2376/1813, 307/518, 536/592, and 690/481 M$^{-1}$cm$^{-1}$, respectively (Supplementary Table 6). Moreover, **R/S-Eu-Et-2** also showed obvious mirror image CPL signal peaks at 580, 609, 635, and 671 nm, which can also be attributed to the $^5D_0 \to {}^7F_{1-4}$ energy level of Eu$^{III}$ ions transition (Fig. 4I). The corresponding $g_{lum}$ values of **R/S-Eu-Et-2** at 575, 619, 635, and 671 nm are 0.06/0.08, 0.02/0.02, 0.16/0.17 and 0.05/0.05, respectively (Fig. 4J). The $B_{CPL}$ values were 121/150, 40/38, 321/320, and 100/94 M$^{-1}$cm$^{-1}$, respectively. Although the maximum $B_{CPL}$ value of chiral Eu$^{III}$ complexes is 3240 M$^{-1}$cm$^{-1}$, the $B_{CPL}$ of most chiral Eu$^{III}$ complexes is still less than 500 M$^{-1}$cm$^{-1}$ (Supplementary Table 7). Therefore, **R/S-Eu-Et-1** with a $B_{CPL}$ value of 2429/2353 M$^{-1}$cm$^{-1}$ is at a high level. In summary, these chiral Eu$^{III}$ complex emitters (**R/S-Eu-R-1** (R = Et/Me) and **R/S-Eu-Et-2**) possess both high $g_{lum}$ and $B_{CPL}$ values (realizing the double improvement of the two), indicating good CPL performance, and suggesting great application prospects in the 3D optical display, biological imaging, chiral optoelectronic devices, and sensors.

## Sensing properties of **R-Eu-Et-1/2**

Compared with high-nuclear lanthanide clusters or polymers (Ln-HOFs, Ln-MOFs, etc.), mononuclear lanthanide complexes have better dispersion and solution processability. Therefore, simple and cheap lanthanide mononuclear complexes are easier to process further and have higher application prospects in the fields of smart sensing, solid-state lighting, and multiple anti-counterfeiting[57]. On the basis of the structural characteristics and excellent luminescent properties of **R-Eu-Et-1/2**, the intelligent sensing of different heavy metal ions dispersed in CH$_3$CN solution was further explored. Specifically, aqueous solutions containing different metal ions (such as Cu$^{II}$, Co$^{II}$, Fe$^{III}$, Ag$^I$, Ca$^{II}$, Cd$^{II}$, Ce$^{III}$, K$^I$, Mg$^{II}$, Mn$^{II}$, Nd$^{III}$, Ni$^{II}$, Pr$^{III}$, Y$^{III}$, and Zn$^{II}$) at the same concentration (20 μM) were added to acetonitrile solutions (3 mL) containing **R-Eu-Et-1** or **R-Eu-Et-2**. Observe the luminescence change of the above mixed solution under a 365 nm UV-lamp. **R-Eu-Et-1** or **R-Eu-Et-2** without additional metal ions showed bright red luminescence. In addition, most of the solutions with added metal ions still maintained bright red luminescence (Fig. 5A, B). However, the red luminescence of **R-Eu-Et-1** solutions containing Cu$^{II}$, Co$^{II}$, and Fe$^{III}$ ions was effectively quenched; the Cu$^{II}$ ions showed the best quenching effect, and the solution luminescence was almost completely lost (Fig. 5C). Similarly, the luminescence of **R-Eu-Et-2** in solutions containing Cu$^{II}$ and Co$^{II}$ ions was effectively quenched (Fig. 5D).

Accordingly, we tested the response sensitivities of **R-Eu-Et-1/2** dispersed in CH$_3$CN solution to different contents of Cu$^{II}$ ions and calculated their limits of detection (LOD). Specifically, 0–6/12 μL of Cu$^{II}$ ion solution (20 μM) was accurately pipetted and added to the CH$_3$CN solution containing 2 mL of **R-Eu-Et-1/2**, respectively, and observed the luminescence change of the above mixed solution under a 365 nm UV-lamp. Solutions containing **R-Eu-Et-1** or **R-Eu-Et-2** exhibited characteristic emissions of Eu$^{III}$ ions ($^5D_0 \to {}^7F_1$, 592 nm, $^5D_0 \to {}^7F_2$, 617 nm, $^5D_0 \to {}^7F_3$, 650 nm, and $^5D_0 \to {}^7F_4$, 685 nm) which gradually decreased with the continuous addition of Cu$^{II}$ ions, and were finally completely quenched (Fig. 5E, F). The red characteristic emissions of the **R-Eu-Et-1/2** solutions were completely quenched when the added Cu$^{II}$ ion solution reached 6 and 9 μL, respectively. According to the Stern–Volmer equation ($I_0/I = 1 + K_{SV}[M]$), the detection limits for Cu$^{II}$ were calculated to be 2.55 and 4.44 nM, respectively, for the **R-Eu-Et-1/2** solutions (Fig. 5G). Note that the detection limits of **R-Eu-Et-1/2** for Cu$^{II}$ ions are much below the maximum approved concentration prescript by the USEPA, evidencing the great potential of **R-Eu-Et-1/2** for the detection of trace amounts of the heavy metal ion Cu$^{II}$ in water. In addition, the UV-Vis absorption spectra of **R-Eu-Et-1/2**, **R-Eu-Et-1** + Cu$^{II}$, and **R-Eu-Et-2** + Cu$^{II}$ were tested respectively, and the results showed that they all showed an obvious absorption peak with a wavelength of approximately 295 nm, which can be attributed to the characteristic absorption of organic ligands (Supplementary Fig. 33). It is worth noting that the above UV-Vis absorption peak does not have obvious characteristic absorption of Cu$^{II}$ ions, indicating that the concentration of Cu$^{II}$ ions in the sensing process is particularly low. We further explored the optical sensing sensitivity of **Eu-Et-3** to Cu$^{II}$ ions (the experimental conditions are the same as **R-Eu-Et-1/2**). Cu$^{II}$ ions sensing experiments show that the detection limit of **R-Eu-Et-1** with AIE behavior is much higher than that of **Eu-Et-3** (LOD = 534.22 nM), which is about 209 times (Supplementary Fig. 34). Therefore, the introduction of the molecular rotor or vibration unit strategy to construct lanthanide complexes with AIE behavior opens up a horizons for improving their optical, sensing and other properties. The capability of efficiently identifying the focal object in a complex environment is an essential feature of a sensor. **R-Eu-Et-1** can efficiently respond to trace amounts of Cu$^{II}$ ions under the conditions of a variety of interfering ions, a feature which proves its reliability as an excellent sensor component (Supplementary Fig. 35). Compared with other Cu$^{II}$ ion sensors, such as ratiometric fluorescent probes (ZTM@FITC, 5.61 nM and 4.96 nM)[38], polyacrylonitrile nanoparticles (tPAN NPs, 10 nM)[58],

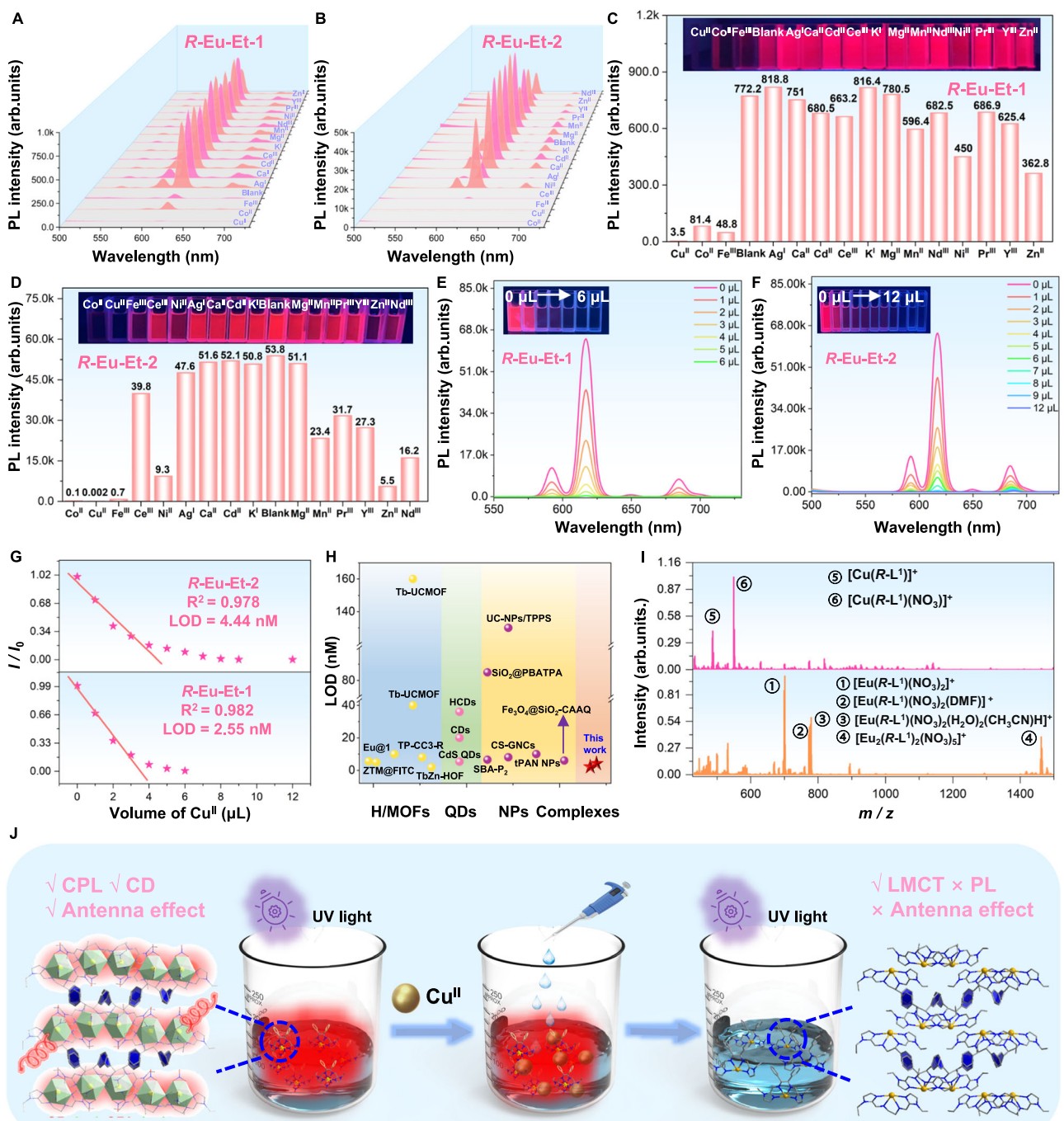

**Fig. 5 | Study on the sensing performance. A, B** Under 365 nm excitation, emission spectra of ***R*-Eu-Et-1/2**. **C, D** Luminescence intensity comparison of ***R*-Eu-Et-1/2** solutions after adding different metal ions (insets in **C**, **D**: photos of ***R*-Eu-Et-1/2** photoresponses to different metal ions under 365 nm UV-light). Concentration-dependent spectra and fitting curves of ***R*-Eu-Et-1/2** on Cu$^{II}$ ions under excitation at 365 nm (**E** and **G** for ***R*-Eu-Et-1**, **F**, and **G** for ***R*-Eu-Et-2**). **H** Performance comparison of ***R*-Eu-Et-1/2** and different types of materials reported for Cu$^{II}$ ion sensing. **I** Molecular ion peaks and analytical results of the HRESI-MS of ***R*-Eu-Et-1** before and after adding Cu$^{II}$ ions. **J** Schematic diagram of the photoresponses of ***R*-Eu-Et-1** or ***R*-Eu-Et-2** to heavy metal ion Cu$^{II}$. Note: The yellow ball represents the series of hydrogen bonds/metal-organic framework; the pink ball represents the series of quantum dots; the purple ball represents the series of nanoparticles; the red star represents this work; The shade of red represents the glow color of ***R*-Eu-Et-1** or ***R*-Eu-Et-2**.

inorganic−organic silica materials (SBA-P2, 6.5 nM)[59], carbon dots (CD, 20 nM)[60], fluorescently active hybrid materials (SiO$_2$@PBATPA, 85 nM and 184 nM)[61], lanthanide-doped metal-organic frameworks (Eu@1, 10 nM)[62], and our previously reported lanthanide hydrogen-bonded organic framework (TbZn-HOF, 1.91 nM)[14], ***R*-Eu-Et-1** and ***R*-Eu-Et-2** have lower detection limits for Cu$^{II}$, with excellent sensing performance (Fig. 5H). In addition, we also explored the optical response sensitivity of ***R*-Eu-Et-1/2** in the aggregated state to Co$^{II}$ and Fe$^{III}$ ions

respectively and obtained their detection limits. First, a Co$^{II}$ ion aqueous solution with a concentration of 0.46 mM was gradually added to the ***R*-Eu-Et-1/2** solution, and it was observed that the characteristic emission of Eu$^{III}$ ions in the above solution gradually weakened and was eventually completely quenched. The LODs of ***R*-Eu-Et-1/2** for Co$^{II}$ ions were calculated to be 53 and 32.8 nM, respectively (Supplementary Fig. 36). Similarly, with the continuous addition of Fe$^{III}$ ion aqueous solution with a concentration of 0.47 mM, the emission peak of ***R*-Eu-

**Et-1/2** gradually weakened and was finally completely quenched. Further, the LODs of **R-Eu-Et-1/2** for $Fe^{III}$ ions were 48 and 71 nM, respectively (Supplementary Fig. 37). The detection of low-concentration metal ions often mainly uses instruments such as inductively coupled plasma mass spectrometry and atomic absorption spectrometry. The above-mentioned detection process usually relies on bulky instruments, complex sampling procedures, and execution by professionals, making the entire detection process cumbersome, time-consuming, and costly. **R-Eu-Et-1/2** with excellent optical properties is expected to be prepared into fluorescent test paper or a convenient identifier for optical sensing of specific metal ions with the help of anti-counterfeiting ink technology. Simple optical sensing test strip technology can effectively avoid the cumbersome procedures and expensive costs of large-scale instrument testing. Target analytes can be detected with the naked eye only with the help of portable low-power UV-lamps. Therefore, **R-Eu-Et-1/2** are expected to become a simple, economical, portable, and highly visible in vitro metal ion optical sensing test strip.

To explore the sensing mechanism of chiral mononuclear $Eu^{III}$ complexes **R-Eu-Et-1/2** to heavy metal ion $Cu^{II}$, the **R-Eu-Et-1** solution before and after adding $Cu^{II}$ ions was subjected to HRESI-MS. In the positive ion mode, the key molecular ion peaks in the **R-Eu-Et-1** solution without adding $Cu^{II}$ ions were $[Eu(R\text{-}L^1)(NO_3)_2]^+$ ($m/z = 701.13$) (1), $[Eu(R\text{-}L^1)(NO_3)_2(DMF)]^+$ ($m/z = 774.19$) (2), $[Eu_2(R\text{-}L^1)(NO_3)_2(H_2O)_2(CH_3CN)H]^+$ ($m/z = 779.14$) (3) and $[Eu_2(R\text{-}L^1)_2(NO_3)_5]^+$ ($m/z = 1426.26$) (4) (Fig. 5I and Supplementary Fig. 38). The main molecular ion peaks in the **R-Eu-Et-1** solution after adding $Cu^{II}$ ions were $[Cu(R\text{-}L^1)]^+$ ($m/z = 487.16$) (5), and $[Cu(R\text{-}L^1)(NO_3)]^+$ ($m/z = 549.15$) (6) (Fig. 5I and Supplementary Fig. 38). The above HRESI-MS results indicate that $Cu^{II}$ ions can replace $Eu^{III}$ ions in the chiral $Eu^{III}$ complex structure, leading to strong LMCT to achieve luminescence quenching (Fig. 5J). In conclusion, both chiral mononuclear $Eu^{III}$ complexes **R-Eu-Et-1/2** exhibit excellent dynamic optical sensing properties for trace amounts of $Cu^{II}$ ions, opening up-to-date horizons for the development of ultra-efficient and sensors for heavy metal ions.

With rapid industrial development, the amount of acid/alkaline wastewater has increased swiftly. Such wastewater contains various harmful substances or heavy metal salts, which is highly corrosive, has seriously threatened human health, and caused irreversible damage to the ecology. Therefore, the development of sensors that accurately monitor pH is of great significance to various research fields ranging from the environment to chemical biology[63]. In recent years, lanthanide complex emitters have been widely used in chemical sensing, such as the detection of explosives, heavy metal ions, and small organic molecules[7,8]. However, the progress of lanthanide complex emitters for efficient light-responsive pH sensors still lags. Therefore, constructing optical sensors of lanthanide complexes with smart photo-responsiveness in a narrow pH range for precise measurement of acidic and basic pH values remains extremely challenging. Based on the aforementioned work and background, we explored the photo-response behavior of chiral $Eu^{III}$ complexes **R-Eu-Et-1/2** in a specific pH range. Specifically, 0–110 μL of the aqueous solution of $C_{(H^+)} = 0.001$ mol$L^{-1}$ (pH = 3) was accurately pipetted into the $CH_3CN$ solution containing **R-Eu-Et-1/2** (2 mL), respectively, and the luminescence changes of the above mixed solutions were observed under the irradiation of a 365 nm UV-light. Solutions containing both **R-Eu-Et-1/2** showed efficient and clear characteristic emission of $Eu^{III}$ ions ($^5D_0 \to {}^7F_1$, 592 nm, $^5D_0 \to {}^7F_2$, 617 nm, $^5D_0 \to {}^7F_3$, 650 nm, and $^5D_0 \to {}^7F_4$, 685 nm). With the continuous addition of pH = 3 acidic aqueous solution, the characteristic emission peak intensity of the chiral lanthanide complexes gradually decreased (Fig. 6A, D). Similarly, accurately pipetted 0–55/65 μL of $C_{(OH^-)} = 0.01$ mol$L^{-1}$ (pH = 12) aqueous solution was introduced into the acetonitrile solution (2 mL) containing **R-Eu-Et-1/2**, and observed the luminescence change of the above mixed solution under the irradiation of 365 nm UV-light. Both solutions containing **R-Eu-Et-1/2** showed the characteristic emission of $Eu^{III}$

ions, and the intensity of the above emission peaks gradually decreased with the continuous addition of pH = 12 aqueous solutions (Fig. 6B, E). The red characteristic emission of **R-Eu-Et-1/2** solutions was completely quenched when the added acid or alkaline solution reached 110 and 55/65 μL, respectively. Note that both chiral $Eu^{III}$ complexes **R-Eu-Et-1/2** achieved a linear decrease in emission in the ranges of pH = 4.28–6.3 and pH = 9.3–11.42. According to the Stern–Volmer equation, their detection limits for the $C_{(H^+)} = 0.001$ mol$L^{-1}$ and $C_{(OH^-)} = 0.01$ mol$L^{-1}$ aqueous solution were 1.91 and 2.84 μM (**R-Eu-Et-1/2**, for $C_{(H^+)}$); 4.44 and 12.58 μM (**R-Eu-Et-1/2**, for $C_{(OH^-)}$) (Fig. 6C, F). Compared with the reported fluorescent pH sensors (example: Zn-MOF, pH = 4–11[64], Zr-MOF, pH = 4.6–7.12[65], NGQDs, pH = 1.8–13.5[66], Tb-NCs, pH = 3–10[67], Tb-MOF, pH = 2–7[68], and Lum-MDI-CA/MTPP, pH = 1–2 and 12–14[63]), **R-Eu-Et-1/2** can intelligently photoresponse to acidic and alkaline solutions in a narrower pH range, and the response process shows a linear change, achieving excellent pH sensing behavior (Fig. 6G). As far as we know, this work is a rare example of the application of chiral lanthanide complex emitters to sensing of low-concentration acidic or alkaline solutions, which provides a strategy for the development of photoresponsive pH sensors with high sensitivity. In addition, the optical response sensitivities of **R-Eu-Et-1/2** to $Cu^{II}$ ions under different pH conditions were explored, and their LODs were obtained. Specifically, $Cu^{II}$ ion aqueous solutions under acidic or alkaline conditions (pH = 4, 5, 6, 9, 10, and 11) were gradually added to solution **R-Eu-Et-1** or **R-Eu-Et-2** respectively, and it was observed that the characteristic emission of $Eu^{III}$ ions of the above solutions gradually weakened and was eventually completely quenched. Through calculation, the LODs of **R-Eu-Et-1/2** for $Cu^{II}$ ions under different pH conditions are 0.081 and 0.65 nM (pH = 4), 0.086 and 0.64 nM (pH = 5), 0.1 and 0.99 nM (pH = 6), 0.11 and 0.86 nM (pH = 9), 0.1 and 0.98 nM (pH = 10), 0.12 and 0.85 nM (pH = 11) (Supplementary Fig. 39, 40). The above experimental results prove that **R-Eu-Et-1/2** can be used to sense trace amounts of heavy metal ions $Cu^{II}$ ions under different pH conditions, showing great potential for specific sensing of $Cu^{II}$ ions in complex environmental pollutants.

## Double anti-counterfeiting performance of R-Eu-Et-1/2

With the rapid development of fluorescent materials, anti-counterfeiting technology is also constantly innovating. Among such technology, anti-counterfeiting fluorescent materials used in photo-luminescent printing have shown great application prospects in the field of anti-counterfeiting given their low cost, simple design, and environmental protection[24,69]. The efficient and fast response behavior of **R-Eu-Et-1/2** to the aqueous solution with low concentration (trace amount) of $Cu^{II}$ ions further expands their application in double anti-counterfeiting. Specifically, we dispersed **R-Eu-Et-1/2** in an aqueous solution with polyethylene glycol and sonicated the mixture to make a simple ink which was then evenly spread on ordinary filter paper with patterns and numbers. Under a daylight lamp, only black and white peony flower patterns and the number 8 can be seen, but under the condition of 365 nm UV-light, the peony flower pattern and the number 8 can be seen with the eye by emitting strong red light (Fig. 7). Note that when the $Cu^{II}$ ion aqueous solution with a concentration of only 6 μM is sprayed on the center of the peony flower and a part of the number 8, the obvious red luminescence can be quickly observed to be completely quenched, which shows the fast and efficient anti-counterfeiting characteristics.

## Discussion

We carefully designed and selected chiral ligands with the molecular rotor or vibrating element structures, and constructed three pairs of dynamic chiral $Eu^{III}$ complex emitters with AIE behavior through the RIR or RIV mechanism in the induced aggregation state **R/S-Eu-R-1** and **R/S-Eu-R-2** (**1**, R = Et/Me; **2**, R = Et). **R/S-Eu-R-1** and **R/S-Eu-R-2** in the molecular state have no obvious luminescence behaviors and exhibit

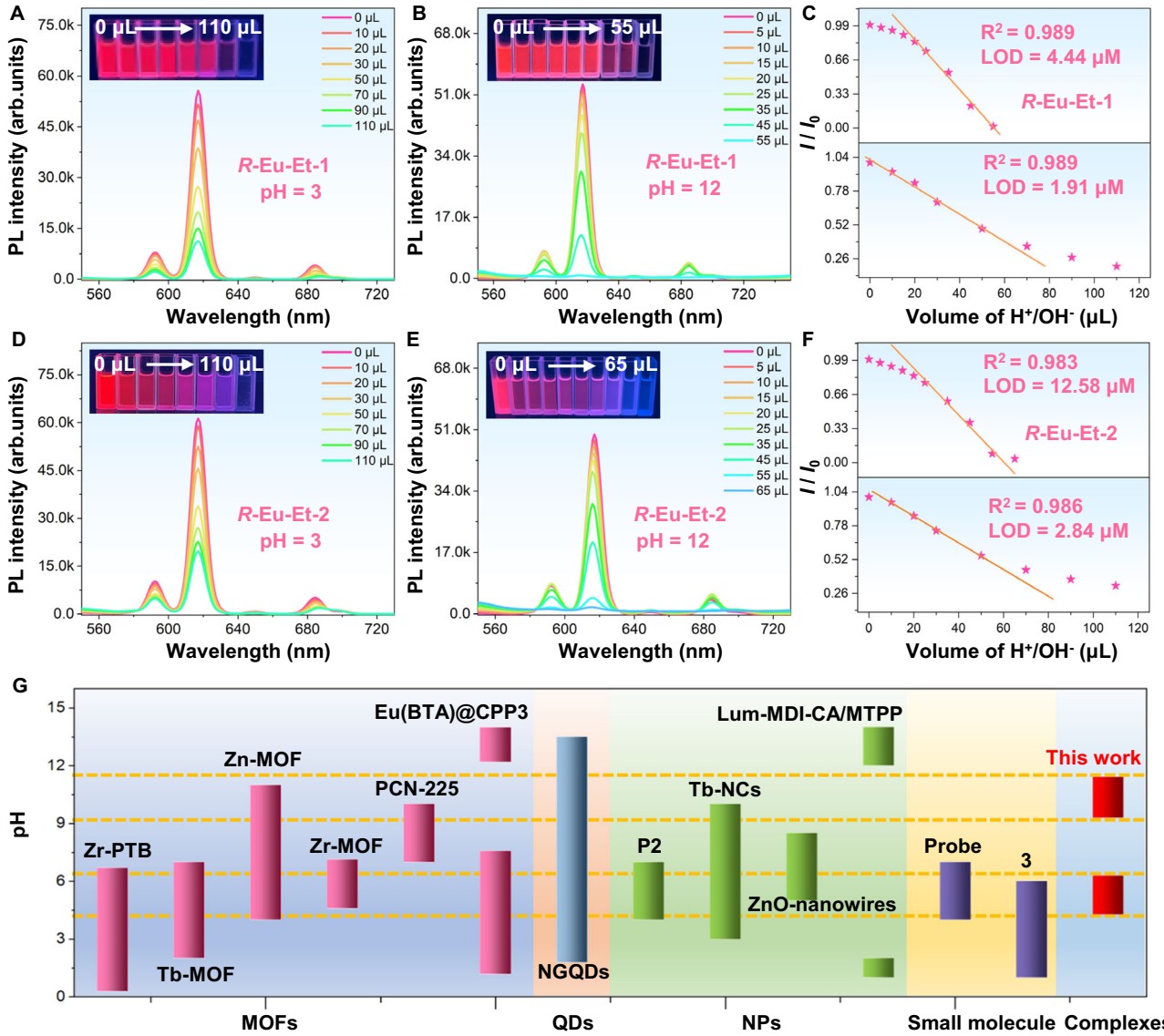

**Fig. 6 | Sensing experiment of acid and alkali solutions.** *R*-Eu-Et-1 to $C_{(H^+)}$ = 0.001 M (pH = 3) (**A**) and $C_{(OH^-)}$ = 0.01 M (pH = 12) (**B**) concentration-dependent PL spectra under excitation at 365 nm and fitting curves of aqueous solutions (**C**). **E, F** *R*-Eu-Et-2 to $C_{(H^+)}$ = 0.001 M (**D**) and $C_{(OH^-)}$ = 0.01 M (**E**) concentration-dependent PL spectra under excitation at 365 nm and fitting curves of aqueous solutions (**F**). **G** Comparison of the pH ranges of *R*-Eu-Et-1/2 and different types of pH sensors that have been reported. Note: Dark blue shading represents a series of metal-organic frameworks; orange shading represents quantum dots; green shading represents a series of nanoparticles; yellow shading represents small molecules; light blue shading represents this work.

the obvious characteristic emission of Eu^III ions; moreover, the emission of such ions increases rapidly with the increase of the aggregation degree. In the molecular state of *R/S*-Eu-R-1 and *R/S*-Eu-Et-2, the molecular rotors or vibrational elements in the structure dissipate the energy of the excited state through non-radiative transitions. In the aggregated state, the movement of the rotors or vibrational elements is restricted, leading to the opening of the ISC process. Further, the aggregation-promoted antenna effect is realized to obtain bright emission. In addition, both *R/S*-Eu-R-1 (R = Et/Me) and *R/S*-Eu-Et-2 showed excellent CPL performances, with the $g_{lum}$ of *R*-Eu-Et-1 as high as 0.64 ($^5D_0 \rightarrow {}^7F_1$) and the $B_{CPL}$ as high as 2429 M$^{-1}$cm$^{-1}$, achieving a rare double increase in the $g_{lum}$ and $B_{CPL}$ values. The construction of organic ligands based on RIR and RIV properties usually requires complex operations and low-yield organic synthesis. We used a simple and efficient solvothermal one-pot method to obtain *R/S*-Eu-R-1 (R = Et/Me) and *R/S*-Eu-Et-2, thereby opening a door for the construction of dynamic chiral lanthanide complex emitters. The metal center Eu^III of *R*-Eu-Et-1/2 with bright emission can be replaced by the heavy metal

ion Cu^II, showing a fast and selective photoresponse to trace Cu^II ions, and their LODs were as low as 2.55 and 4.44 nM, significantly outperforming the other types of luminescent materials. In addition, *R*-Eu-Et-1 can selectively photoresponse to acidic or basic aqueous solutions in a narrow pH range, showing a rare high sensitivity to low-content acidic and basic solutions. Finally, this work introduces chiral molecular rotors or vibrational units to provide vivid examples of the construction of dynamic chiral lanthanide complexes with bright emission. These chiral Eu^III complexes have high CPL parameters and sensing properties, opening a horizon for the construction of multifunctional chiral lanthanide complex emitters.

## Methods

### The synthesis method

**Synthesis of [Eu(*R*-L¹)(NO₃)₃] (*R*-Eu-Et-1).** Add 0.1 mmol (0.0456 g) Eu(NO₃)₃·6H₂O, 0.1 mmol (0.0212 g) (1 *R*,2 *R*)-(-)-diphenylethylenediamine, 0.1 mmol (0.0124 g) 1-ethyl-1H-imidazole-2-carbaldehyde and mixed solvent (EtOH:CH₃CN = 1:1) were added in a pyrex tube (22 cm).

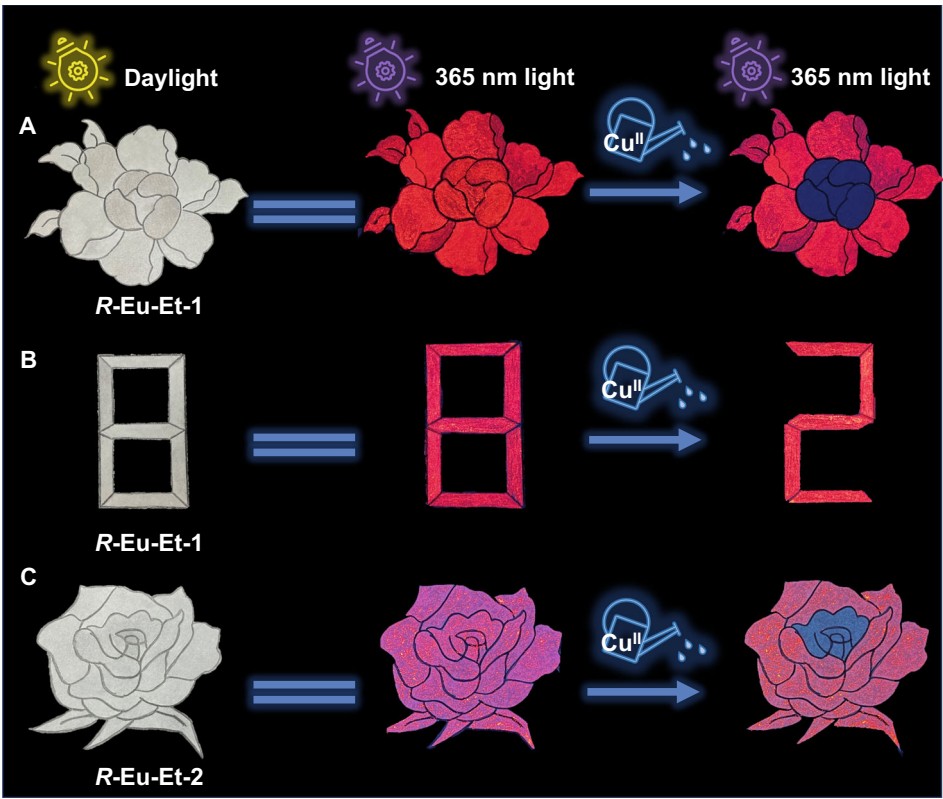

**Fig. 7 | Anti-counterfeiting performance.** Patterns of peony flowers and the number 8 drawn by anti-counterfeit inks containing *R*-Eu-Et-1 (**A**, **B**) and *R*-Eu-Et-2 (**C**) under a daylight lamp and at 365 nm and after spraying the Cu$^{II}$ ion solution security pattern.

In the tube, shake and sonicate for 10 min. Place the sealed pyrex tube in an oven at 80 °C, take it out one days later, slowly cool to room temperature, and precipitate yellow lumpy crystals. The yield is about 51% (calculated with the amount of Eu(NO$_3$)$_3$·6H$_2$O). Elemental analysis theoretical value (C$_{26}$H$_{28}$EuN$_9$O$_9$): C, 40.95%; H, 3.70%; N, 16.53%; experimental value: C, 40.88%; H, 3.67%; N, 16.50%. Infrared spectrum data (IR, KBr pellet, cm$^{-1}$): 3422(m), 3127(m), 2980(w), 2494(s), 2312(s), 1628(s), 1472(s), 1376(s), 1160(m), 986(s), 778(s), 614(s), 536(s).

**Synthesis of [Eu(S-L$^1$)(NO$_3$)$_3$] (S-Eu-Et-1).** Add 0.1 mmol (0.0456 g) Eu(NO$_3$)$_3$·6H$_2$O, 0.1 mmol (0.0212 g) (1 S,2 S)-(+)-diphenylethylene-diamine, 0.1 mmol (0.0124 g) 1-ethyl-1H-imidazole-2-carbaldehyde and mixed solvent (EtOH:CH$_3$CN = 1:1) were added in a pyrex tube (22 cm). In the tube, shake and sonicate for 10 min. Place the sealed pyrex tube in an oven at 80 °C, take it out one days later, slowly cool to room temperature, and precipitate yellow lumpy crystals. The yield is about 48.5% (calculated with the amount of Eu(NO$_3$)$_3$·6H$_2$O). Elemental analysis theoretical value (C$_{26}$H$_{28}$EuN$_9$O$_9$): C, 40.95%; H, 3.70%; N, 16.53%; experimental value: C, 40.90%; H, 3.64%; N, 16.48%. Infrared spectrum data (IR, KBr pellet, cm$^{-1}$): 3430(m), 3120(m), 2980(w), 2494(s), 1628(s), 1484(s), 1382(m), 1165(m), 983(m), 784(m), 619(s), 532(s).

**Synthesis of [Eu(R-L$^1$)(NO$_3$)$_3$] (R-Eu-Me-1).** Add 0.1 mmol (0.0456 g) Eu(NO$_3$)$_3$·6H$_2$O, 0.1 mmol (0.0212 g) (1 R,2 R)-(-)-diphenylethylenedia-mine, 0.1 mmol (0.011 g) 1-methyl-1H-imidazole-2-carbaldehyde and mixed solvent (EtOH:CH$_3$CN = 1:1) were added in a pyrex tube (22 cm). In the tube, shake and sonicate for 10 min. Place the sealed pyrex tube in an oven at 80 °C, take it out one days later, slowly cool to room temperature, and precipitate yellow lumpy crystals. The yield is about 44.9% (calculated with the amount of Eu(NO$_3$)$_3$·6H$_2$O). Elemental analysis theoretical value (C$_{24}$H$_{24}$EuN$_9$O$_9$): C, 39.25%; H, 3.29%; N, 17.16%; experimental value: C, 39.17%; H, 3.24%; N, 17.11%. Infrared

spectrum data (IR, KBr pellet, cm$^{-1}$): 3376(s), 2948(w), 1633(s), 1492(s), 1383(s), 1301(s), 1165(m), 1024(m), 952(w), 860(m), 774(m), 619(w), 532(w).

**Synthesis of [Eu(S-L$^1$)(NO$_3$)$_3$] (S-Eu-Me-1).** Add 0.1 mmol (0.0456 g) Eu(NO$_3$)$_3$·6H$_2$O, 0.1 mmol (0.0212 g) (1 S,2 S)-(+)-diphenylethylene-mine, 0.1 mmol (0.011 g) 1-methyl-1H-imidazole-2-carbaldehyde and mixed solvent (EtOH:CH$_3$CN = 1:1) were added in a pyrex tube (22 cm). In the tube, shake and sonicate for 10 min. Place the sealed pyrex tube in an oven at 80 °C, take it out one days later, slowly cool to room temperature, and precipitate yellow lumpy crystals. The yield is about 39.7% (calculated with the amount of Eu(NO$_3$)$_3$·6H$_2$O). Elemental analysis theoretical value (C$_{24}$H$_{24}$EuN$_9$O$_9$): C, 39.25%; H, 3.29%; N, 17.16%; experimental value: C, 39.19%; H, 3.22%; N, 17.13%. Infrared spectrum data (IR, KBr pellet, cm$^{-1}$): 3395(s), 2947(w), 1637(m), 1488(s), 1378(m), 1300(s), 1025(m), 957(m), 856(w), 764(w), 619(w), 533(w).

**Synthesis of [Eu(R-L$^2$)(NO$_3$)$_3$]·CH$_3$CN (R-Eu-Et-2).** Add 0.1 mmol (0.0456 g) Eu(NO$_3$)$_3$·6H$_2$O, 0.1 mmol (0.0114 g) (1 R,2 R)-(-)-1,2-diami-nocyclohexane, 0.1 mmol (0.0124 g) 1-ethyl-1H-imidazole-2-carbalde-hyde and mixed solvent (EtOH:CH$_3$CN = 1:1) were added in a pyrex tube (22 cm). In the tube, shake and sonicate for 10 min. Place the sealed pyrex tube in an oven at 80 °C, take it out one days later, slowly cool to room temperature, and precipitate yellow lumpy crystals. The yield is about 45.7% (calculated with the amount of Eu(NO$_3$)$_3$·6H$_2$O). Elemental analysis theoretical value (C$_{20}$H$_{29}$EuN$_{10}$O$_9$): C, 34.05%; H, 4.14%; N, 19.85%; experimental value: C, 33.99%; H, 4.11%; N, 19.75%. Infrared spectrum data (IR, KBr pellet, cm$^{-1}$): 3430(m), 2942(w), 1628(m), 1460(w), 1381(s), 1025(w), 857(w), 619(m).

**Synthesis of [Eu(S-L$^2$)(NO$_3$)$_3$]·CH$_3$CN (S-Eu-Et-2).** Add 0.1 mmol (0.0456 g) Eu(NO$_3$)$_3$·6H$_2$O, 0.1 mmol (0.0114 g) (1 S,2 S)-( + )-1,2-diami-nocyclohexane, 0.1 mmol (0.0124 g) 1-ethyl-1H-imidazole-2-

carbaldehyde and mixed solvent (EtOH:CH$_3$CN = 1:1) were added in a pyrex tube (22 cm). In the tube, shake and sonicate for 10 min. Place the sealed pyrex tube in an oven at 80 °C, take it out one days later, slowly cool to room temperature, and precipitate yellow lumpy crystals. The yield is about 46.3% (calculated with the amount of Eu(NO$_3$)$_3$·6H$_2$O). Elemental analysis theoretical value (C$_{20}$H$_{29}$EuN$_{10}$O$_9$): C, 34.05%; H, 4.14%; N, 19.85%; experimental value: C, 34.01%; H, 4.09%; N, 19.79%. Infrared spectrum data (IR, KBr pellet, cm$^{-1}$): 3416(m), 2942(w), 1627(m), 1460(w), 1383(s), 1025(w), 855(w), 619(m).

**Synthesis of [Eu(L$^3$)(NO$_3$)$_3$] (Eu-Et-3).** Add 0.1 mmol (0.0456 g) Eu(NO$_3$)$_3$·6H$_2$O, 0.1 mmol (0.006 g) ethylenediamine, 0.1 mmol (0.0124 g) 1-ethyl-1H-imidazole-2-carbaldehyde and mixed solvent (EtOH:CH$_3$CN = 1:1) were added in a pyrex tube (22 cm). In the tube, shake and sonicate for 10 min. Place the sealed pyrex tube in an oven at 80 °C, take it out one days later, slowly cool to room temperature, and precipitate white lumpy crystals. The yield is about 56% (calculated with the amount of Eu(NO$_3$)$_3$·6H$_2$O). Elemental analysis theoretical value (C$_{14}$H$_{20}$EuN$_9$O$_9$): C, 27.55%; H, 3.30%; N, 20.65%; experimental value: C, 27.52%; H, 3.28%; N, 20.62%.

**Synthesis of [Gd(R/S-L$^1$)(NO$_3$)$_3$] (R/S-Gd-Et-1).** Add 0.1 mmol (0.0451 g) Gd(NO$_3$)$_3$·6H$_2$O, 0.1 mmol (0.0212 g) (1 R/S,2 R/S)-(-/+)-diphenylethylenediamine, 0.1 mmol (0.0124 g) 1-ethyl-1H-imidazole-2-carbaldehyde and mixed solvent (EtOH:CH$_3$CN = 1:1) were added in a pyrex tube (22 cm). In the tube, shake and sonicate for 10 min. Place the sealed pyrex tube in an oven at 80 °C, take it out one days later, slowly cool to room temperature, and precipitate yellow lumpy crystals. The yield is about 46.5% (calculated with the amount of Gd(NO$_3$)$_3$·6H$_2$O). Elemental analysis theoretical value (C$_{26}$H$_{28}$N$_9$O$_9$Gd): C, 40.67%; H, 3.68%; N, 16.42%; experimental value: C, 40.60%; H, 3.65%; N, 16.41%.

**Synthesis of [Gd(R/S-L$^2$)(NO$_3$)$_3$]·CH$_3$CN (R/S-Gd-Et-2).** Add 0.1 mmol (0.0451 g) Gd(NO$_3$)$_3$·6H$_2$O, 0.1 mmol (0.0114 g) (1 R/S,2 R/S)-(-/+)-1,2-diaminocyclohexane, 0.1 mmol (0.0124 g) 1-ethyl-1H-imidazole-2-carbaldehyde and mixed solvent (EtOH:CH$_3$CN = 1:1) were added in a pyrex tube (22 cm). In the tube, shake and sonicate for 10 min. Place the sealed pyrex tube in an oven at 80 °C, take it out one days later, slowly cool to room temperature, and precipitate yellow lumpy crystals. The yield is about 48.5% (calculated with the amount of Gd(NO$_3$)$_3$·6H$_2$O). Elemental analysis theoretical value (C$_{20}$H$_{29}$GdN$_{10}$O$_9$): C, 33.80%; H, 4.11%; N, 19.71%; experimental value: C, 33.72%; H, 4.04%; N, 19.66%.

**Synthesis of [Gd(R/S-L$^1$)(NO$_3$)$_3$] (R/S-Gd-Me-1).** Add 0.1 mmol (0.0451 g) Gd(NO$_3$)$_3$·6H$_2$O, 0.1 mmol (0.0212 g) (1 R/S,2 R/S)-(-/+)-diphenylethylenediamine, 0.1 mmol (0.011 g) 1-methyl-1H-imidazole-2-carbaldehyde and mixed solvent (EtOH:CH$_3$CN = 1:1) were added in a pyrex tube (22 cm). In the tube, shake and sonicate for 10 min. Place the sealed pyrex tube in an oven at 80 °C, take it out one days later, slowly cool to room temperature, and precipitate yellow lumpy crystals. The yield is about 45.9% (calculated with the amount of Gd(NO$_3$)$_3$·6H$_2$O). Elemental analysis theoretical value (C$_{24}$H$_{24}$GdN$_9$O$_9$): C, 38.97%; H, 3.27%; N, 17.04%; experimental value: C, 38.91%; H, 3.25%; N, 16.98%.

**Synthesis of [Tb(R/S-L$^1$)(NO$_3$)$_3$] (R/S-Tb-Et-1).** Add 0.1 mmol (0.0453 g) Tb(NO$_3$)$_3$·6H$_2$O, 0.1 mmol (0.0212 g) (1 R/S,2 R/S)-(-/+)-diphenylethylenediamine, 0.1 mmol (0.0124 g) 1-ethyl-1H-imidazole-2-carbaldehyde and mixed solvent (EtOH:CH$_3$CN = 1:1) were added in a pyrex tube (22 cm). In the tube, shake and sonicate for 10 min. Place the sealed pyrex tube in an oven at 80 °C, take it out one days later, slowly cool to room temperature, and precipitate yellow lumpy crystals. The yield is about 46.5% (calculated with the amount of Tb(NO$_3$)$_3$·6H$_2$O). Elemental analysis theoretical value (C$_{26}$H$_{28}$N$_9$O$_9$Tb): C, 40.58%; H, 3.67%; N, 16.38%; experimental value: C, 40.52%; H, 3.63%; N, 16.31%.

**Synthesis of [Tb(R/S-L$^2$)(NO$_3$)$_3$]·CH$_3$CN (R/S-Tb-Et-2).** Add 0.1 mmol (0.0453 g) Tb(NO$_3$)$_3$·6H$_2$O, 0.1 mmol (0.0114 g) (1 R/S,2 R/S)-(-/ + )-1,2-diaminocyclohexane, 0.1 mmol (0.0124 g) 1-ethyl-1H-imidazole-2-carbaldehyde and mixed solvent (EtOH:CH$_3$CN = 1:1) were added in a pyrex tube (22 cm). In the tube, shake and sonicate for 10 min. Place the sealed pyrex tube in an oven at 80 °C, take it out one days later, slowly cool to room temperature, and precipitate yellow lumpy crystals. The yield is about 50.5% (calculated with the amount of Tb(NO$_3$)$_3$·6H$_2$O). Elemental analysis theoretical value (C$_{20}$H$_{29}$TbN$_{10}$O$_9$): C, 33.72%; H, 4.10%; N, 19.66%; experimental value: C, 33.69%; H, 4.06%; N, 19.56%.

**Synthesis of [Tb(R/S-L$^1$)(NO$_3$)$_3$] (R/S-Tb-Me-1).** Add 0.1 mmol (0.0453 g) Tb(NO$_3$)$_3$·6H$_2$O, 0.1 mmol (0.0212 g) (1 R/S,2 R/S)-(-/+)-diphenylethylenediamine, 0.1 mmol (0.011 g) 1-methyl-1H-imidazole-2-carbaldehyde and mixed solvent (EtOH:CH$_3$CN = 1:1) were added in a pyrex tube (22 cm). In the tube, shake and sonicate for 10 min. Place the sealed pyrex tube in an oven at 80 °C, take it out one days later, slowly cool to room temperature, and precipitate yellow lumpy crystals. The yield is about 52.1% (calculated with the amount of Tb(NO$_3$)$_3$·6H$_2$O). Elemental analysis theoretical value (C$_{24}$H$_{24}$TbN$_9$O$_9$): C, 38.88%; H, 3.26%; N, 17.00%; experimental value: C, 38.83%; H, 3.22%; N, 16.94%.

## Data availability

The authors declare that all other data supporting the findings of this study are provided in the Supplementary Information/Source Data file. Source data are provided with this paper. All other data are available from the corresponding author upon request. The X-ray crystallographic coordinates for structures reported in this study have been deposited at the Cambridge Crystallographic Data Center (CCDC), under deposition numbers are 2262497 (**R-Eu-Et-1**), 2262500 (**S-Eu-Et-1**); 2262499 (**R-Eu-Me-1**), 2262503 (**S-Eu-Me-1**); 2262498 (**R-Eu-Et-2**), 2262502 (**S-Eu-Et-2**); 2307112 (**Eu-Et-3**); 2308025 and 2308037 (**R/S-Gd-Et-1**); 2308026 (**R-Gd-Et-2**); 2308027 and 2308038 (**R/S-Gd-Me-1**); 2308028 and 2308029 (**R-Tb-Et-1/2**); 2308033 and 2308040 (**R/S-Tb-Me-1**). These data can be obtained free of charge from The Cambridge Crystallographic Data Center via www.ccdc.cam.ac.uk/data_request/cif. The atomic coordinates of structures for DFT calculation are provided as a Source Data file. The experimental data used in this study are available in the open repository Figshare under accession code https://doi.org/10.6084/m9.figshare.24763749. Source data are provided with this paper.

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

## Acknowledgements

This work was supported by the National Natural Science Foundation of China (NSFC 22061005 from H.-H. Z, 22270168 from F.-P. L, and 22075058 from F.-P. L).

## Author contributions

Y. L. and H. W. tested, collected and analyzed data for all complexes. Y. L. writing-original draft. Y. W. and Y. L. synthesized complexes. Z. Z. conceived the ain idea of this work and revised original draft. H. Z. and F. L. provided financial support; H. Z. and Z. Z. supervised and revised the manuscript. All authors discussed the results and commented on the manuscript.

## Competing interests

The authors declare no competing interests.
