## [Peer Review File · Nature Communications]

Aggregation Induced Emission Dynamic Chiral Europium(III) Complexes with Excellent Circularly Polarized Luminescence and Smart SensorsREVIEWER COMMENTS

Reviewer #1 (Remarks to the Author):

The manuscript by Zou reported an example of chiral Eu-complexes with CPL emission, and try to show their applications in sensor and Anti-counterfeit. After carefully reading the manuscript, this reviewer found the novel, content, especially the essential understanding to luminescence theory is far from the standard of NC for the high-quality paper. Two main issues need discuss: 1) the authors misuse the AIE concept to luminescent Ln complexes. For lanthanide complexes, the triplet energy match is the most important factor to influence the luminescent. This reviewer believes the enhancement of luminescence for the complexes arise from the decrease of T1 state of L, although the contributions from molecular vibration and rotation are involved. This T1 variation can be deduced from the bathochromic-shift of absorbance shown in Figure S8. For a molecule with AIE feature, such as TPE, the aggregation will not cause the spectral shift, other words, the shift of energy level. 2) For a sensor, the sensing substrate can interact with probe by chemistry reaction or intermolecular interactions, but not to destroy the integrated structure of probe molecule skeleton. In this work, the replacement of Eu by Cu obviously destroyed the structure to form a new complex. Thus, the description of mechanism for the decrease of emission by using word “quenching” is not suitable, actually it is “destroy”. By this way, any substrates that can destroy the complex’s structure all will response to this luminescent quenching. Actually, for some color ions, such as Cu²⁺, Fe³⁺, they generally will cause the quenching of emission for almost any luminophores, because there are absorbance competitions between color ion and organic molecule or ligand.

Additionally, some words are exaggeratory or unsuitable, such as “Intelligent Sensing”, “Multi-Smart Sensors”. As a scientific paper, accurately expressing the scientific focus of a work by simple language is more conducive the readers to understand the highlight of a paper, rather than try to make up “a colorful story”. There are many issues that exist in the manuscript, I think they needn’t to be listed here due to their relatively far distance to the acceptance by high-level journals, even those of modest-level journals.

Reviewer #2 (Remarks to the Author):

Zou and co-workers used chiral organic ligands with molecular rotor structures or vibrational units to react with Eu^{III} ion under solvothermal “one-pot” conditions to obtain a series of dynamic chiral europium complexes that can exhibit AIE behavior. The authors tested the AIE titration curves, quantum yields, and lifetimes of these compounds, and analyzed the differences between the aggregation state and the molecular state in combination with the single crystal structure. Notably, R/S-Eu-R-1 has both high glum and BCPL. Obtaining chiral lanthanide complexes with both high glum and BCPL values is a very encouraging result. In addition, the metal-centered Eu^{III} ions of R-Eu-Et-1/2 can be replaced by heavy metal ion Cu^I, thus achieving a highly sensitive smart photoresponsive to Cu^I. And R-Eu-Et-1 can

also achieve highly sensitive response to acidic and alkaline aqueous solutions. Overall, this manuscript provides a facile and efficient strategy for constructing dynamic chiral lanthanide complexes with bright emission. Besides that, the manuscript is creative and well organized with sufficient data. It is believed that this manuscript can be published on Nature Communications after minor revisions.

1. For the convenience of the reader, the authors should give detailed annotations of “5D0→7F1, 5D0→7F2, 5D0→7F3, and 5D0→7F4”, especially the definition of these specific excited states, in the legend of Figure 3A.
2. It is recommended that the author add atom labels in Figure 1, so that readers can obtain relevant information faster and understand the composition of the complex.
3. Similarly, the corresponding complexes and other relevant information should be indicated in Figures 2, 4 and 5.
4. The authors should add the Zeta potential to demonstrate the stability of the aggregates in solution.
5. Chiral europium complexes have obvious photoresponse effects on heavy metal ions CuII, CoII and FeII. Why did the author only provide the LOD and selective experimental results of CuII ions. The results of the LOD and selectivity experiments of the smart photoresponse CoII and FeII ions should be further provided and analyzed.
6. The UV-Vis absorption spectra before and after the smart response to CuII ions are missing.
7. The results of the manuscript’s detection of heavy metal ions have been well explained and supported by relevant evidence. However, I think that at the end of the discussion the authors could add a short paragraph to highlight their findings. In short, their findings can be updated by simple comparisons with conventional techniques or with reported detection modalities.
8. Among the many rare earth metal ions with excellent luminescent properties, why did the author choose europium? For other rare earth metals, such as Sm and Tb, Do they also have the same excellent luminescence and CPL properties as the complexes reported in this manuscript.
9. In the Supporting Information, the author only puts the SEM images of R-Eu- Et/Me-1 and R/S-Eu-Et-2, what about other complexes?
10. The BCPL value of R/S-Eu-Et-1 is 2429/2353 M⁻¹cm⁻¹. Compared with all the complexes reported so far, what level is the BCPL value of R/S-Eu-Et-1? What is the highest BCPL value for chiral europium complexes? If the author can provide such a comparison, I believe it will better illustrate the “R/S-Eu-Et-1 with excellent CPL performance” described by the author.

Reviewer #3 (Remarks to the Author):

This manuscript introduces a series of innovative chiral dynamic luminescent complexes of Eu(III), referred to as R/SEu-R-1/2. These complexes exhibit distinctive characteristics upon aggregation. In their molecular state, their luminescence is minimal; however, when aggregated, their emission intensities undergo significant enhancement. This phenomenon is attributed to the constraint on molecular rotor or vibrational element motion, resulting in improved intersystem crossing efficiency. Moreover, the R/SEu-R-1/2 complexes demonstrate very interesting circularly polarized luminescence (CPL) performance,

especially R-S-Eu-R-1 that stands out with remarkable emission performance quantified by high asymmetry factors and brightness values. The complexes also display versatile sensing capabilities, showcasing remarkable sensitivity in detecting trace amounts of Cu(II) ions, with detection limits as low as 2.55-4.44 nM. Finally, the application of the complexes extends to rapid anti-counterfeiting through photoluminescent printing (the compounds emit red light when exposed to 365 nm UV-light, which can be rapidly quenched upon exposure to Cu(II) ions).

The results are interesting with potential to catch the attention of the journal readers. Nevertheless, the discussion of the photophysical properties of the prepared materials require a significant revision (see details below). Furthermore, and regarding the proposed applications, it is unclear the advantages of the presented approach and materials with respect to what was already reported.

Weaknesses

1) In Figures 2B,E the emission intensity is discussed using the comparison with the value recorded at 0% acetonitrile fraction. Although frequently used in the literature, caution must be taken when comparing intensities, as the differences observed may be attributed to variations in the absorption coefficient among the samples. To draw quantitative conclusions, it is essential to measure the emission quantum yield of the different samples.

2) Correct the units for absorbance in Figures 2C and 2F. Absorbance (or optical density) is a nondimensional quantity defined as the logarithm of the intensity ratio of the beam passing through the sample and by a reference path. Moreover, ensure that the reference beam is an empty cuvette when reporting the absorbance of mixed solutions.

3) Elaborate on the depicted energy transfer paths in Figure 2G, providing support from numerical simulations or relevant literature.

4) Consider moving the motivation for detecting Cu(II) to the introduction section to make the text more concise in the "Multiple Intelligent Sensing Properties" section.

5) Clarify the specificity of the sensor to Cu(II) ions and its ability to distinguish from other species. Explain the horizontal axis on the limit of detection plot in Figure 4H.

6) Evaluate the dependence of sensor performance on pH values in the presence of Cu(II) ions and report the findings.

7) The figures' quality in the PDF file provided for review is low and difficult to read. Higher-resolution figures should be included for clarity and ease of interpretation.

8) Reduce the use of unnecessary adjectives like "intelligent" or "smart" and instead use more precise descriptors.

9) Clarify the meaning of the horizontal axis in Figure 5G.

10) Explain the main advantages of the proposed anti-counterfeiting approach compared to other existing technologies. Make the wetting of the sensor with a Cu(II)-containing solution clearer. Maintain consistent nomenclature as Cu(II) and remove Cu²⁺ from the figure.

11) Address whether the sensor's response is reversible after coming into contact with Cu(II) ions in the "Double Anti-counterfeiting Performance" section.

Responses to Reviewers:

Reviewer 1:

The manuscript by Zou reported an example of chiral Eu-complexes with CPL emission, and try to show their applications in sensor and anti-counterfeit. After carefully reading the manuscript, this reviewer found the novel content, especially the essential understanding to luminescence theory is far from the standard of *Nature Communications* for the high-quality paper.

Reply: Thank you very much for taking the time out of your busy schedule to review this manuscript and for providing your valuable suggestions for this manuscript. Based on your suggestions, we have made significant revisions to the manuscript. We believe that the manuscript after major revisions can resolve most of your doubts, and the quality of the manuscript has been greatly improved.

Two main issues need discuss:

1) the authors misuse the AIE concept to luminescent Ln complexes. For lanthanide complexes, the triplet energy match is the most important factor to influence the luminescent. This reviewer believes the enhancement of luminescence for the complexes arise from the decrease of T₁ state of L, although the contributions from molecular vibration and rotation are involved. This T₁ variation can be deduced from the bathochromic-shift of absorbance shown in Figure S8. For a molecule with AIE feature, such as TPE, the aggregation will not cause the spectral shift, other words, the shift of energy level.

Reply: Many thanks to the reviewer for his comments. In this manuscript we have not misused the AIE concept to explain lanthanide complex emitters. As for the absorbance in Figure S8 (Figure 15 in the Revised Supplementary Information), there has been a certain degree of red shift (*R/S-Eu-Me-1* red-shifted 23 and 21 nm respectively; *S-Eu-Et-1/2* red-shifted 18 and 22 nm respectively). Attributed to the large difference in size in the aggregated and dispersed states of these lanthanide complexes. In addition, in order to further prove the AIE behaviors of these lanthanide complex emitters, we further added the following experiments:

First, we changed the metal center Eu^{III} ions to Tb^{III}/Gd^{III} ions to obtain *R/S-Gd-R-1* (**R = Et/Me**), *R/S-Tb-R-1*, *R/S-Gd-Et-2*, and *R/S-Tb-Et-2*. Due to the energy level mismatch between

the organic ligands and Gd^{III} ions and Tb^{III} ions, **R/S-Gd-R-1** (R = Et/Me), **R/S-Tb-R-1**, **R/S-Gd-Et-2**, and **R/S-Tb-Et-2** all only showed the luminescence of the ligands. Similarly, we dissolved equal amounts of **R/S-Gd-Et-1**, **R/S-Tb-Et-1**, **R-Gd-Me-1**, **R-Tb-Me-1**, **R/S-Gd-Et-2**, and **R/S-Tb-Et-2** in glycerol/DMSO mixed solutions with different proportions and tested the emission spectra of the above solutions. Experimental results show that **R/S-Gd-Et-1**, **R/S-Tb-Et-1**, **R-Gd-Me-1**, **R-Tb-Me-1**, **R/S-Gd-Et-2**, and **R/S-Tb-Et-2** all exhibit obvious AIE behaviors.

In addition, we supplemented the particle sizes of **R/S-Eu-Me-1** and **S-Eu-Et-1/2** respectively in the aggregated state. The experimental results show that the particle sizes of **R/S-Eu-Me-1** aggregates in CH₃CN/DMF mixed solution with a content of 99% are 712/615 nm respectively; and the particle sizes of **S-Eu-Et-1/2** aggregates in CH₃CN/DMF mixed solution with a content of 99% are 390/396 nm, respectively. Therefore, their absorbance undergoes a certain degree of red shift, which can be attributed to the large difference in size between the aggregated and dispersed states of these lanthanide complexes. It is not caused by the decrease in the energy level of the T₁ state of the organic ligand.

Finally, according to a series of reported works, it is a normal phenomenon that the UV-Vis absorption of complexes with AIE behavior in the molecular state and aggregated state has a certain degree of red shift. For example:

- 1 Y. J. Kong, Z. P. Yan, S. Li, H. F. Su, K. Li, Y. X. Zheng, S. Q. Zang, *Angew. Chem. Int. Ed.* **2020**, *59*, 5336–5340.
- 2 M. Zhang, X. Dong, Z. Wang, H. Li, S. Li, X. Zhao, S. Zang, *Angew. Chem. Int. Ed.* **2019**, *131*, 2–8.
- 3 Y. Tian, Z. Y. Wang, S. Q. Zang, D. Li, T. C. W. Mak, *Dalton. Trans.* **2019**, *48*, 2275–2279.
- 4 Y. Y. Liu, X. Zhang, K. Li, Q. C. Peng, Y. J. Qin, H. W. Hou, S. Q. Zang, B. Z. Tang, *Angew. Chem. Int. Ed.* **2021**, *60*, 22417–22423.
- 5 H. F. Su, Q. C. Peng, Y. U. Liu, T. Xie, P. P. Liu, Y. C. Cai, W. Wen, Y. H. Yu, K. Li, S. Q. Zang, *Biomaterials* **2022**, *288*, 121691.
- 6 Y. Jin, Q. Peng, S. Li, H. Su, P. Luo, M. Yang, X. Zhang, K. Li, S. Zang, B. Z. Tang, T. C. W. Mak, *Natl Sci Rev.* **2022**, *9*, nwab216.
- 7 Y. Jin, Q. C. Peng, J. W. Xie, K. Li, S. Q. Zang, *Angew. Chem. Int. Ed.* **2023**, *62*, e202301000.

8 P. P. Dong, Y. Y. Liu, Q. C. Peng, H. Y. Li, K. Li, S. Q. Zang, B. Z. Tang, *Dalton. Trans.* **2023**, 52, 1913–1918.

Regarding the red shift of the absorption spectrum caused by the increase in particle size, please refer to the following references:

- 1 P. Parsamian, Y. Liu, C. Xie, Z. Chen, P. Kang, Y. H. Wijesundara, N. M. Al-Kharji, R. N. Ehrman, O. Trashi, J. Randrianalisoa, X. Zhu, M. D'Souza, L. A. Wilson, M. J. Kim, Z. Qin, J. J. Gassensmith, *ACS Nano* **2023**, 17, 7797–7805.
- 2 T. Kim, Q. Zhang, J. Li, L. Zhang, J. V. Jokerst, *ACS Nano* **2018**, 12, 5615–5625.
- 3 T. T. S. Lew, K. M. M. Aung, S. Y. Ow, S. N. Amrun, L. Sutarlie, L. F. P. Ng, X. Su, *ACS Nano* **2021**, 15, 12286–12297.
- 4 K. Salorinne, R. W. Y. Man, C. H. Li, M. Taki, M. Nambo, C. M. Crudden, *Angew. Chem. Int. Ed.* **2017**, 56, 6198–6202.
- 5 Z. Jia, M. Ben Amar, D. Yang, O. Brinza, A. Kanaev, X. Duten, A. Vega-González, *Chem. Eng. J.* **2018**, 347, 913–922.
- 6 Y. Dou, Y. Guo, X. Li, X. Li, S. Wang, L. Wang, G. Lv, X. Zhang, H. Wang, X. Gong, J. Chang, *ACS Nano* **2016**, 10, 2536–2548.

Taken together, these evidences are sufficient to demonstrate that these lanthanide complexes exhibit significant AIE behavior and are not responsible for the luminescence changes due to the reduction of the T₁ state of the organic ligands.

2) For a sensor, the sensing substrate can interact with probe by chemistry reaction or intermolecular interactions, but not to destroy the integrated structure of probe molecule skeleton. In this work, the replacement of Eu by Cu obviously destroyed the structure to form a new complex. Thus, the description of mechanism for the decrease of emission by using word “quenching” is not suitable, actually it is “destroy”. By this way, any substrates that can destroy the complex's structure all will response to this luminescent quenching. Actually, for some color ions, such as Cu²⁺, Fe³⁺, they generally will cause the quenching of emission for almost any luminophores, because there are absorbance competitions between color ion and organic molecule or ligand.

Reply: Thank you for your suggestion. In this manuscript, the luminescence quenching behavior of **R-Eu-Et-1** caused by Cu^{II} ions can be called “static quenching”. We screened a large number of references and found that many well-known research groups also called “the behavior in which the luminescence of the sensor gradually decreases until it disappears during the sensing process” as “quenching”. For example:

- Wang *et al.* clearly stated in the title “*Tuning the Luminescence of Metal–Organic Frameworks for Detection of Energetic Heterocyclic Compounds*”: “When Mg²⁺ ions are replaced by Ni²⁺ and Co²⁺ respectively to form the other two MOFs, Φ_F is effectively reduced, especially Co-MOF, which has no visible light emission under 365 nm UV lamp irradiation. The results show that the LMCT transition controls the intrinsic optical properties of both MOFs, thereby quenching their fluorescence.” on *J. Am. Chem. Soc.* **2014**, *136*, 15485–15488;
- Zeng *et al.* clearly stated in the title “*Real-time and visual sensing devices based on pH-control assembled lanthanide-barium nano-cluster*”: “In the exploration of the sensing mechanism, the disappearance of Tb₂Ba₂ luminescence in strong acid and alkali solutions can be attributed to the dissociation of nanoclusters. The above phenomenon is called luminescence quenching.” on *J. Hazard. Mater.* **2021**, *413*, 125291;
- Wang *et al.* clearly stated in the title “*Fluorescence and Colorimetric Dual-Mode Ratiometric Sensor Based on Zr–Tetraphenylporphyrin Tetrasulfonic Acid Hydrate Metal–Organic Frameworks for Visual Detection of Copper Ions*”: “In the sensing experiments, it may be because Cu²⁺ may combine with the amino groups in ZTMs to form a non-fluorescent amino complex (Cu²⁺-TPPS). Therefore, Cu²⁺ is an important factor leading to FL quenching.” on *ACS Appl. Mater. Interfaces* **2022**, *14*, 13848–13857;
- Zeng *et al.* clearly stated in the title “*Synthesis of Two New Dinuclear Lanthanide Clusters and Visual Bifunctional Sensing Devices Based on the Eu Cluster*”: “Research on the sensing mechanism shows that due to the special d²sp³ hybridization of Fe²⁺ providing a pair of electrons to the empty orbit of dmp, a Fe²⁺-dmp coordination complex may eventually be formed. Therefore, Fe²⁺ quenches the fluorescence of 1-Eu.” on *Adv. Optical Mater.* **2022**, *10*, 2102267;

- Moonhyun Oh *et al.* clearly stated in the title “*Dual Changes in Conformation and Optical Properties of Fluorophores within a Metal–Organic Framework during Framework Construction and Associated Sensing Event*”: “Fluorescence titration of CPP-16 with Cu²⁺ in MeCN. As the amount of Cu²⁺ increases, the fluorescence intensity of the CPP-16 pyrene excimer emission band decreases. Therefore, the fluorescence quenching of CPP-16 is due to the interaction between Cu²⁺ and CPP-16.” on *J. Am. Chem. Soc.* **2014**, *136*, 12201–12204;
- Peter J. Stang *et al.* clearly stated in the title “*Multicomponent Platinum(II) Cages with Tunable Emission and Amino Acid Sensing*”: “In experiments examining amino acids, the amino acids serve as better ligands for Pt(II) ions, leading to the disruption of the cage and the formation of mononuclear Pt(II)-amino acid complexes. Emitting ligand 5 is then released, which serves as an indicator for sensing thiol-containing amino acids.” on *J. Am. Chem. Soc.* **2017**, *139*, 5067–5074;
- Balouch *et al.* clearly stated in the title “*Pyranine functionalized Fe₃O₄ nanoparticles for the sensitive fluorescence detection of Cu²⁺ ions*”: “A fluorophore molecule forms a stable complex with another atom or molecule (the quencher) that does not fluoresce in the ground state and results in a quenching called static quenching.” on *J. Alloys Compd.* **2018**, *767*, 151-162;
- Zhao *et al.* clearly stated in the title “*Tailoring the Fluorescence of AIE-Active Metal–Organic Frameworks for Aqueous Sensing of Metal Ions*”: “The fluorescence off state of Zn(HTABDC)(Bpy)·DMF is explained by the fluorescence quenching mechanism based on the light-induced electron transfer from the excited state of the linker HTABDC to Bpy.” on *ACS Appl. Mater. Interfaces* **2018**, *10*, 3801–3809.

Therefore, we believe that the behavior in which **R-Eu-Et-1** light responds to Cu^{II} ions and causes its luminescence to gradually weaken until it disappears can also be called “quenching”. In addition, the following references also use “quenching” to describe the behavior of luminescence weakening:

- 1 S. Chatterjee, H. Gohil, I. Raval, S. Chatterjee, A. R. Paital, *Small* **2019**, *15*, 1804749.
- 2 S. Wu, Y. Lin, J. Liu, W. Shi, G. Yang, P. Cheng, *Adv. Funct. Mater.* **2018**, *28*, 1707169.
- 3 Q. Meng, X. Zhang, C. He, G. He, P. Zhou, C. Duan, *Adv. Funct. Mater.* **2010**, *20*, 1903–1909.
- 4 X. Guo, N. Zhu, S.P. Wang, G. Li, F.Q. Bai, Y. Li, Y. Han, B. Zou, X. B. Chen, Z. Shi, S.

Feng, *Angew. Chem. Int. Ed.* **2020**, *59*, 19716–19721.

Although Cu^{II} or Fe^{III} ions are non-ferrous metal ions, it is obvious that they only have an easily distinguishable and obvious color under high concentration conditions, and do not require sensing to quickly identify them. However, the heavy metal ions often used in sensing experiments are of extremely low concentration and cannot be effectively distinguished by the naked eye. Especially for Cu^{II} ions, which are toxic to heavy metals, their color is negligible at low concentrations. Therefore, it is very important to develop fast and highly sensitive sensing for extremely low concentrations of Cu^{II} ions. In addition, we screened a large amount of literature and found that many well-known research groups have also explored sensing work on non-ferrous metal ions. For example:

- 1 W. Cho, H. J. Lee, G. Choi, S. Choi, M. Oh, *J. Am. Chem. Soc.* **2014**, *136*, 12201–12204.
- 2 L. Basabe-Desmonts, J. Beld, R. S. Zimmerman, J. Hernando, P. Mela, M. F. García Parajó, N. F. Van Hulst, A. Van Den Berg, D. N. Reinhoudt, M. Crego-Calama, *J. Am. Chem. Soc.* **2004**, *126*, 7293–7299.
- 3 Q. Meng, X. Zhang, C. He, G. He, P. Zhou, C. Duan, *Adv. Funct. Mater.* **2010**, *20*, 1903–1909.
- 4 S. Chatterjee, H. Gohil, I. Raval, S. Chatterjee, A. R. Paital, *Small* **2019**, *15*, 1–16.
- 5 I. Lee, S. Kim, S. N. Kim, Y. Jang, J. Jang, *ACS Appl. Mater. Interfaces* **2014**, *6*, 17151–17156.

3. Additionally, some words are exaggeratory or unsuitable, such as “Intelligent Sensing”, “Multi-Smart Sensors”. As a scientific paper, accurately expressing the scientific focus of a work by simple language is more conducive to the readers to understand the highlight of a paper, rather than try to make up “a colorful story”. There are many issues that exist in the manuscript, I think they needn’t to be listed here due to their relatively far distance to the acceptance by high-level journals, even those of modest-level journals.

Reply: Thank you for your suggestion. Based on your suggestions, we have carefully checked and deleted exaggerated and inappropriate related descriptions in the manuscript, such as: multi-smart sensing, etc. We have used simple language to accurately express the scientific ideas of this manuscript. Furthermore, “smart sensing” has been widely used and therefore, “smart sensing” is still retained in this manuscript.

- The progress entitled “*Functional Coatings on High-Performance Polymer Fibers for Smart Sensing*” by Hegemann *et al.* (*Adv. Funct. Mater.* **2020**, *30*, 1910555);
- The progress entitled “*Rational Fabrication of Crystalline Smart Materials for Rapid Detection and Efficient Removal of Ozone*” by Zhang *et al.* (*Angew. Chem. Int. Ed.*, **2021**, *60*, 6055–6060);
- The progress entitled “*Recent advances in covalent organic frameworks (COFs) as a smart sensing material*” by Chen *et al.* (*Chem. Soc. Rev.*, **2019**, *48*, 5266-5302);
- The progress entitled “*A double responsive smart upconversion fluorescence sensing material for glycoprotein*” by Wang *et al.* (*Biosens. Bioelectron.* **2016**, *85*, 596–602);
- The progress entitled “*Protein-Inspired Self-Healable Ti₃C₂ MXenes/ Rubber-Based Supramolecular Elastomer for Intelligent Sensing*” by Lu *et al.* (*ACS Nano* **2020**, *14*, 2788–2797);
- The progress entitled “*A signal-amplified ratiometric fluorescence biomimetic sensor based on the synergistic effect of IFE and AE for the visual smart monitoring of oxytetracycline*” by Wang *et al.* (*Chem. Eng. J.* **2022**, *433*, 134499);
- The progress entitled “*Polymeric Schiff base assisted synthesis of Fe-N-C MFs single-atom nanozymes for discrimination and intelligent sensing of tannic acid*” by Li *et al.* (*Chem. Eng. J.* **2023**, *468*, 143638).

In addition, in order to improve the quality of this manuscript, resolve the doubts raised by reviewers, and make it more in line with the interests of *Nature Communications* readers, we have further supplemented the following experimental results:

- ◆ By changing (1*R*,2*R*)-1,2-Diphenylethylenediamine to ethylenediamine under the same synthesis conditions, a lanthanide complex **Eu-Et-3** without rotors and vibrational elements and with significant ACQ behavior was successfully obtained. Comparative experimental results show that introducing molecular rotors or vibration units into lanthanide complexes can not only effectively increase the solid-state QYs (increased by 9.76 and 8.57 times respectively), but also greatly improve the optical sensing sensitivity of copper ions (increased 209.49 times). Therefore, we believe that introducing organic ligands with molecular rotors or

vibration units into lanthanide complexes can effectively promote the improvement of optical properties, sensing and other properties of lanthanide complexes.

- ◆ In addition, ***R/S-Gd-R-1*** (**R = Et/Me**), ***R/S-Tb-R-1***, ***R/S-Gd-Et-2***, and ***R/S-Tb-Et-2*** were obtained by only replacing the metal center Eu^{III} with $\text{Tb}^{\text{III}}/\text{Gd}^{\text{III}}$ ions under the same conditions as ***R-Eu-Et-1***. The emission spectra of the above lanthanide complexes were tested in mixed solutions of glycerol/DMSO with different ratios (f_w), and the results showed that they all exhibited obvious AIE behavior. Therefore, these evidences are sufficient to prove that these lanthanide complexes have significant AIE behavior, and are not due to the luminescence changes caused by the reduction of the T_1 state of the organic ligands.
- ◆ In addition, the absorption peaks of these lanthanide complexes in the molecular state and the aggregated state ($f_w = 0\%$ and $f_w = 99\%$) have a certain degree of red shift, which is caused by changes in their sizes in the two different states. We have provided professional references and provided DLS results to prove our claims.
- ◆ The quantum yields of ***R/S-Eu-Et-1***, ***R/S-Eu-Et-2***, and ***R/S-Eu-Me-1*** in mixed solutions of glycerol/DMSO or $\text{CH}_3\text{CN}/\text{DMF}$ at different ratios (f_w) were further supplemented. Experimental results show that as the proportion of glycerol or CH_3CN continues to increase, the quantum yields of these lanthanide complexes continue to increase, once again proving that they have aggregation-enhanced antenna effect behavior.
- ◆ Finally, in order to further understand and explore the ***R/S-Eu-Et-1*** and ***R/S-Eu-Et-2*** energy transfer pathway, we provide the energy values of the S_1 state, T_1 state of the ligand, and the lowest excited state of the metal ion. Further detail the energy transfer paths depicted in the Jablonski energy level diagram.

Finally, we believe that the manuscript after major revision has solved the doubts of the editor and reviewers and can arouse the interest of *Nature Communications* readers.

Apart from this, we would like to reiterate again the highlights of this manuscript:

1. Chiral organic ligands with molecular rotor structures or vibration units reacted with lanthanide metal ions under solvothermal conditions, and three pairs of dynamic chiral Eu^{III} complexes with RIR and RIV properties were obtained for the first time. This simple and efficient atomic utilization of solvothermal “one-pot” method effectively avoids the complex synthesis process

and provides a new strategy for the rational construction of dynamic rare earth complex emitters.

2. The chiral Eu^{III} complex **R/S-Eu-R-1** has excellent CPL performance, with high asymmetry factor (g_{lum}) and CPL brightness (B_{CPL}) as high as 0.64 and $2439 \text{ M}^{-1}\text{cm}^{-1}$ respectively, achieving the rare “double promotion of g_{lum} and B_{CPL} ”.
3. The metal center Eu^{III} of the aggregated **R-Eu-Et-1/2** can be replaced by the heavy metal ion Cu^{II} , showing a highly sensitive and photoresponse with LODs as low as 2.55 and 4.44 nM respectively. It is well below the maximum concentration of Cu^{II} ions in drinking water specified by USEPA.
4. Furthermore, **R-Eu-Et-1** exhibits excellent photoresponse to acidic and alkaline aqueous solutions selectively within a narrow pH range.
5. This work introduces chiral molecular rotors or vibration units to construct chiral Eu^{III} complexes with high CPL parameters and sensing properties, which opens up new horizons for the construction of multifunctional chiral lanthanide complex emitters.

Reviewer 2:

Zou and co-workers used chiral organic ligands with molecular rotor structures or vibrational units to react with Eu^{III} ion under solvothermal “one-pot” conditions to obtain a series of dynamic chiral europium complexes that can exhibit AIE behavior. The authors tested the AIE titration curves, quantum yields, and lifetimes of these compounds, and analyzed the differences between the aggregation state and the molecular state in combination with the single crystal structure. Notably, **R/S-Eu-R-1** has both high g_{lum} and B_{CPL} . Obtaining chiral lanthanide complexes with both high g_{lum} and B_{CPL} values is a very encouraging result. In addition, the metal-centered Eu^{III} ions of **R-Eu-Et-1/2** can be replaced by heavy metal ion Cu^{II} , thus achieving a highly sensitive smart photoresponsive to Cu^{II} . And **R-Eu-Et-1** can also achieve highly sensitive response to acidic and alkaline aqueous solutions. Overall, this manuscript provides a facile and efficient strategy for constructing dynamic chiral lanthanide complexes with bright emission. Besides that, the manuscript is creative and well organized with sufficient data. It is believed that this manuscript can be published on *Nature Communications* after minor revisions.

Reply: We sincerely thank the reviewers for recognizing our contributions and providing professional comments on our work. As you are concerned, there are a few issues that need to be addressed. Based on your kind suggestions, we have revised the previous manuscript, as detailed below.

1. For the convenience of the reader, the authors should give detailed annotations of “ $^5D_0 \rightarrow ^7F_1$, $^5D_0 \rightarrow ^7F_2$, $^5D_0 \rightarrow ^7F_3$, and $^5D_0 \rightarrow ^7F_4$ ”, especially the definition of these specific excited states, in the legend of Figure 3A.

Reply: Thank you for your suggestion. We have annotated D and F in the legend to Figure 4A in the revised manuscript. The following description “ D represents the quintet; F represents the septet” has been added to the legend of Figure 4A of the revised manuscript and is highlighted in blue.

2. It is recommended that the author add atom labels in Figure 1, so that readers can obtain relevant information faster and understand the composition of the complex.

Reply: Thank you for your suggestion. We have added these to Figure 1 (Figure 2 in the revised manuscript) and added the latest Figure 1 (Figure 2 in the revised manuscript) to the revised manuscript.

3. Similarly, the corresponding complexes and other relevant information should be indicated in Figures 2, 4 and 5.

Reply: Thank you very much for your suggestion. We have refined the information in Figures 2, 4, and 5 (Figures 3, 5, and 6 in the revised manuscript), and added the latest Figures 2, 4, and 5 (Figures 3, 5, and 6 in the revised manuscript) to the revised manuscript.

4. The authors should add the Zeta potential to demonstrate the stability of the aggregates in solution.

Reply: Thank you for your suggestion. We have tested the Zeta potential of ***R/S-Eu-Et-1***, ***R/S-Eu-Et-2***, and ***R/S-Eu-Me-1*** in the aggregated state respectively. These results indicate that ***R/S-Eu-Et-1***, ***R/S-Eu-Et-2***, and ***R/S-Eu-Me-1*** can all remain stable in the aggregated state. Zeta

potential results have been added to Figure 16 in the Supplementary Information. Additionally, the following description: “In addition, the Zeta potentials of *R/S-Eu-Et-1*, *R/S-Eu-Et-2*, and *R/S-Eu-Me-1* in the aggregated state are $-21.4/-24.23$, $-16.93/-20.13$, and $-21.63/-21.06$ mV, respectively, indicating that their aggregates can remain stable (Supplementary Fig. 16).” has been added to **“Aggregation-induced Emission of *R/S-Eu-R-1* (R = Et/Me) and *R/S-Eu-Et-2*”** of the revised manuscript and is highlighted in blue.

5. Chiral europium complexes have obvious photoresponse effects on heavy metal ions Cu^{II} , Co^{II} and Fe^{III} . Why did the author only provide the LOD and selective experimental results of Cu^{II} ions. The results of the LOD and selectivity experiments of the smart photoresponse Co^{II} and Fe^{III} ions should be further provided and analyzed.

Reply: Thank you for your suggestion. We have supplemented the experimental results of optical sensing titration of *R-Eu-Et-1/2* on Co^{II} and Fe^{III} ions respectively, and obtained the LOD. Relevant results have been added to the **“Results”** section of the revised manuscript and Figures 36 and 37 in the Supplementary Information. The following describes: “In addition, we also explored the optical response sensitivity of *R-Eu-Et-1/2* in the aggregated state to Co^{II} and Fe^{III} ions respectively, and obtained their detection limits (LOD). First, a Co^{II} ion aqueous solution with a concentration of 0.46 mM was gradually added to the *R-Eu-Et-1/2* solution, and it was observed that the characteristic emission of Eu^{III} ions in the above solution gradually weakened and was eventually completely quenched. The LODs of *R-Eu-Et-1/2* for Co^{II} ions were calculated to be 53 and 32.8 nM, respectively (Supplementary Fig. 36). Similarly, with the continuous addition of Fe^{III} ion aqueous solution with a concentration of 0.47 mM, the emission peak of *R-Eu-Et-1/2* gradually weakened and was finally completely quenched. Further, the LODs of *R-Eu-Et-1/2* for Fe^{III} ions were 48 and 71 nM, respectively (Supplementary Fig. 37).” has been added to section **“Sensing Properties of *R-Eu-Et-1/2*”** of the revised manuscript and is highlighted in blue.

6. The UV-Vis absorption spectra before and after the smart response to Cu^{II} ions are missing.

Reply: Thank you for your suggestion. We have added the UV-visible absorption spectra of *R-Eu-Et-1/2*, *R-Eu-Et-1+Cu^{II}*, and *R-Eu-Et-2+Cu^{II}*, and the relevant results have been added to

Figure 33 in the Supplementary Information. The following describes: “In addition, the UV-Vis absorption spectra of **R-Eu-Et-1/2**, **R-Eu-Et-1+Cu^{II}**, and **R-Eu-Et-2+Cu^{II}** were tested respectively, and the results showed that they all showed an obvious absorption peak with a wavelength of approximately 295 nm, which can be attributed to the characteristic absorption of organic ligands (Supplementary Fig. 33). It is worth noting that the above UV-Vis absorption peak does not have obvious characteristic absorption of Cu^{II} ions, indicating that the concentration of Cu^{II} ions in the sensing process is extremely low.” has been added to the **“Sensing Properties of R-Eu-Et-1/2”** section of the revised manuscript and is highlighted in blue.

7. The results of the manuscript’s detection of heavy metal ions have been well explained and supported by relevant evidence. However, I think that at the end of the discussion the authors could add a short paragraph to highlight their findings. In short, their findings can be updated by simple comparisons with conventional techniques or with reported detection modalities.

Reply: Thank you for your suggestion. The following description: “The detection of low-concentration metal ions often mainly uses instruments such as inductively coupled plasma mass spectrometry and atomic absorption spectrometry. The above-mentioned detection process usually relies on bulky instruments, complex sampling procedures, and execution by professionals, making the entire detection process cumbersome, time-consuming, and costly. **R-Eu-Et-1/2** with excellent optical properties is expected to be prepared into fluorescent test paper or a convenient identifier for optical sensing of specific metal ions with the help of anti-counterfeiting ink technology. Simple optical sensing test strip technology can effectively avoid the cumbersome procedures and expensive costs of large-scale instrument testing. Target analytes can be detected with the naked eye only with the help of portable low-power ultraviolet lamps. Therefore, **R-Eu-Et-1/2** are expected to become a simple, economical, portable, and highly visible in vitro metal ion optical sensing test strip.” has been added to section **“Sensing Properties of R-Eu-Et-1/2”** of the revised manuscript and is highlighted in blue.

8. Among the many rare earth metal ions with excellent luminescent properties, why did the author choose europium? For other rare earth metals, such as Sm and Tb, Do they also have the same excellent luminescence and CPL properties as the complexes reported in this manuscript.

Reply: Thank you for your suggestion. First, the emission spectra of ***R/S-Gd-R-1*** (***R = Et/Me***), ***R/S-Tb-R-1***, ***R/S-Gd-Et-2***, and ***R/S-Tb-Et-2*** only show the emission peaks of the ligands, indicating that the organic ligands ***R-L¹*** and ***R-L²*** have poor energy level matching with Tb^{III} and Gd^{III} and cannot perform effective antenna effects. In addition, the energy values of the excited singlet state S₁ and the excited triplet state T₁ of the organic ligands ***R-L¹*** and ***R-L²*** were obtained respectively, and combined with the Jablonski energy level diagram (Fig. 4C), it can be seen that the T₁ state energy level of the organic ligands ***R-L¹*** and ***R-L²*** matches the lowest excited state energy level of the metal ion Eu^{III} to a high degree. It shows that the organic ligands ***R-L¹*** and ***R-L²*** can serve as efficient antennas to promote the emission of Eu^{III}. Finally, among all lanthanide metal ions, the ⁵D₀→⁷F₁ transition of the Eu^{III} ion is the only one with a “g” value within the D^I range, resulting in the Eu^{III} complex having by far the largest luminescence asymmetry factor (*g*_{lum}). Selecting Eu^{III} ions can also be expected to obtain more excellent circularly polarized luminescent materials. Therefore, in this manuscript we chose the Eu^{III} ion.

9. In the Supplementary Information, the author only puts the SEM images of ***R-Eu-Et/Me-1*** and ***R/S-Eu-Et-2***, what about other complexes?

Reply: Thank you for your suggestion. We have supplementally tested the SEM of ***S-Eu-Et-1*** and ***S-Eu-Me-1*** and have added it to Figure 5 in the Supplementary Information. The following description “The SEM images of ***R/S-Eu-Me-1***, ***R/S-Eu-Et-1***, and ***R/S-Eu-Et-2*** clearly show that they are all bulk crystals and the surfaces are very clean (Supplementary Fig. 5).” has been added to section “**Crystal Structural Analysis of *R/S-Eu-R-1* (*R = Et/Me*) and *R/S-Eu-Et-2***” of the revised manuscript and is highlighted in blue.

10. The *B*_{CPL} value of ***R/S-Eu-Et-1*** is 2439/2363 M⁻¹cm⁻¹. Compared with all the complexes reported so far, what level is the *B*_{CPL} value of ***R/S-Eu-Et-1***? What is the highest *B*_{CPL} value for

chiral europium complexes? If the author can provide such a comparison, I believe it will better illustrate the “***R/S-Eu-Et-1*** with excellent CPL performance” described by the author.

Reply: Thank you for your suggestion. We have reviewed extensive work and added a statistical table of B_{CPL} values for chiral Eu^{III} complexes in Table S7 in the Supplementary Information. So far, the maximum B_{CPL} value of the Eu^{III} complex is $3240 \text{ M}^{-1}\text{cm}^{-1}$ (*J. Am. Chem. Soc.* **2019**, *141*, 19634), while the B_{CPL} value of ***R/S-Eu-Et-1*** is $2439 / 2363 \text{ M}^{-1}\text{cm}^{-1}$, which is at a high level.

Reviewer 3:

This manuscript introduces a series of innovative chiral dynamic luminescent complexes of $\text{Eu}(\text{III})$, referred to as ***R/S-Eu-R-1/2***. These complexes exhibit distinctive characteristics upon aggregation. In their molecular state, their luminescence is minimal; however, when aggregated, their emission intensities undergo significant enhancement. This phenomenon is attributed to the constraint on molecular rotor or vibrational element motion, resulting in improved intersystem crossing efficiency. Moreover, the ***R/S-Eu-R-1/2*** complexes demonstrate very interesting circularly polarized luminescence (CPL) performance, especially ***R/S-Eu-R-1*** that stands out with remarkable emission performance quantified by high asymmetry factors and brightness values. The complexes also display versatile sensing capabilities, showcasing remarkable sensitivity in detecting trace amounts of $\text{Cu}(\text{II})$ ions, with detection limits as low as 2.55-4.44 nM. Finally, the application of the complexes extends to rapid anti-counterfeiting through photoluminescent printing (the compounds emit red light when exposed to 365 nm UV-light, which can be rapidly quenched upon exposure to $\text{Cu}(\text{II})$ ions).

The results are interesting with potential to catch the attention of the journal readers. Nevertheless, the discussion of the photophysical properties of the prepared materials require a significant revision (see details below). Furthermore, and regarding the proposed applications, it is unclear the advantages of the presented approach and materials with respect to what was already reported.

Reply: Thank you very much for taking the time out of your busy schedule to read this manuscript carefully and for providing us with a series of constructive suggestions for our work. Based on your suggestions, we have made significant revisions and discussed in detail the photophysical properties

of the as-prepared materials in the manuscript. We believe that the revised version can address your concerns, greatly improve the quality of the manuscript, and is suitable for publication in the high-level and high-impact *Nature Communications*.

1) In Figures 2B, E the emission intensity is discussed using the comparison with the value recorded at 0% acetonitrile fraction. Although frequently used in the literature, caution must be taken when comparing intensities, as the differences observed may be attributed to variations in the absorption coefficient among the samples. To draw quantitative conclusions, it is essential to measure the emission quantum yield of the different samples.

Reply: Thank you for your suggestion. We have additionally tested the luminescence quantum yields of ***R/S-Eu-Et-1***, ***R/S-Eu-Et-2*** and ***R/S-Eu-Me-1*** in mixed solutions of glycerol/DMSO or CH₃CN/DMF at different ratios (f_w). Experimental results show that as the proportion of glycerol or CH₃CN continues to increase, the quantum yields of these Eu^{III} complexes continue to increase, once again proving that they have significant AIE behavior (Table R1). Relevant results have been added to Figures 7-9 in the Supplementary Information.

Additionally, the following description: “To further verify the AIE behavior of ***R/S-Eu-Et-1***, ***R/S-Eu-Et-2***, and ***R/S-Eu-Me-1***, we tested the QYs of ***R/S-Eu-Et-1***, ***R/S-Eu-Et-2***, and ***R/S-Eu-Me-1*** in glycerol/DMSO or CH₃CN/DMF mixed solutions with different ratios. Experimental results show that as the proportion of poor solvent glycerol or CH₃CN continues to increase, the luminescence QY of these Eu^{III} complexes continues to increase. When the glycerol or CH₃CN content is 99%, the QYs of ***R/S-Eu-Et-1***, ***R/S-Eu-Et-2***, and ***R/S-Eu-Me-1*** reach the maximum, which are 27.2/27.9, 27.8/24.1, 25.0/28.0, 26.3/26.7, 18.6/19.9, and 29.5/29.9, respectively, once again proving that they have significant AIE behavior (Supplementary Fig. 7-9).” has been added to section “**Aggregation-induced Emission of *R/S-Eu-R-1* (R = Et/Me) and *R/S-Eu-Et-2***” of the revised manuscript and is highlighted in blue.

Table R1. Quantum yields of ***R/S-Eu-Et-1***, ***R/S-Eu-Et-2***, and ***R/S-Eu-Me-1*** in glycerol/DMSO or CH₃CN/DMF mixed solutions with different ratios (f_w).

	R-Eu-Et-1	S-Eu-Et-1	R-Eu-Et-2	S-Eu-Et-2	R-Eu-Me-1	S-Eu-Me-1
--	-------------------------	-------------------------	-------------------------	-------------------------	-------------------------	-------------------------

Glycerol: DMSO (%)	QY (%)	QY (%)	QY (%)	QY (%)	QY (%)	QY (%)
0	0.1	0.1	0.04	0.02	0.1	0.1
10	0.2	0.2	0.1	0.07	0.1	0.2
20	0.2	0.2	0.3	0.02	0.2	0.2
30	0.4	0.3	0.3	0.02	0.2	0.2
40	0.4	0.4	0.4	0.02	0.4	0.6
50	0.6	1.4	0.6	0.4	0.4	0.2
60	1.7	2.1	2.8	1.3	2.5	1.7
70	1.8	3.8	8.9	8.1	4.5	4.5
80	11.5	12.7	11.1	13.5	18.5	21.5
90	15.3	16.9	19.7	19.0	21.3	23.2
99	27.2	27.9	27.8	24.1	25.0	28.0
	R-Eu-Et-1	S-Eu-Et-1	R-Eu-Et-2	S-Eu-Et-2	R-Eu-Me-1	S-Eu-Me-1
CH ₃ CN:DMF (%)	QY (%)	QY (%)	QY (%)	QY (%)	QY (%)	QY (%)
0	2.4	4.7	3.9	5.6	3.4	3.9
10	7.5	8.4	9.5	10.4	5.4	8.8
20	14.6	14.8	12.1	12.3	6.6	11.8
30	15.5	14.7	12.9	13.7	9.5	14.8
40	16.1	16.9	13.8	14.2	16.7	17.3
50	17.2	17.7	14.1	14.8	19.7	19.5
60	18.8	19.2	14.8	15.4	19.9	23
70	19.9	20.4	15.8	15.5	23.8	19.7
80	20.5	20.6	16	16.7	18.4	24.2
90	22	22.6	16.7	17.2	24.7	26.3
99	26.3	26.7	18.6	19.9	29.5	29.9

2) Correct the units for absorbance in Figures 2C and 2F. Absorbance (or optical density) is a nondimensional quantity defined as the logarithm of the intensity ratio of the beam passing through the sample and by a reference path. Moreover, ensure that the reference beam is an empty cuvette when reporting the absorbance of mixed solutions.

Reply: Thank you for your suggestion. We have corrected the absorbance unit involved in Figure 2 (Figure 3 in the revised manuscript) of the manuscript, changing “Absorbance (a.u.)” to “Absorbance”. In addition, we have subtracted the blank before testing the UV-Vis absorption spectrum of the mixed solution. Finally, based on the reported series of work, it was once again confirmed that the unit of absorbance is “Absorbance”. For example:

1. M. Zhang, X. Dong, Z. Wang, H. Li, S. Li, X. Zhao, S. Zang, *Angew. Chem. Int. Ed.* **2019**, *131*, 2–8.
2. Y. Y. Liu, X. Zhang, K. Li, Q. C. Peng, Y. J. Qin, H. W. Hou, S. Q. Zang, B. Z. Tang, *Angew. Chem. Int. Ed.* **2021**, *60*, 22417–22423.
3. H. F. Su, Q. C. Peng, Y. U. Liu, T. Xie, P. P. Liu, Y. C. Cai, W. Wen, Y. H. Yu, K. Li, S. Q. Zang, *Biomaterials* **2022**, *288*, 121691.
4. P. P. Dong, Y. Y. Liu, Q. C. Peng, H. Y. Li, K. Li, S. Q. Zang, B. Z. Tang, *Dalton Trans.* **2023**, *52*, 1913–1918.
5. Y. Jin, Q. C. Peng, J. W. Xie, K. Li, S. Q. Zang, *Angew. Chem. Int. Ed.* **2023**, *62*, e20301000.

3) Elaborate on the depicted energy transfer paths in Figure 2G, providing support from numerical simulations or relevant literature.

Reply: Thank you very much for your suggestion. We have provided the detailed energy transfer pathway for **R-Eu-Et-1/2** and the corresponding energy values in Figure 4C of the manuscript. Specifically, according to the ultraviolet absorption-diffuse reflection spectra of the ligands (**R-L¹** and **R-L²**), the energy values of the excited singlet states S₁/S₁' of **R-L¹** and **R-L²** were obtained: 27174 and 28735 cm⁻¹ respectively. In addition, **R-Gd-Et-1/2** were synthesized, and their phosphorescence spectra were tested at 77 K. The energy values of the excited triplet states T₁/T₁' of **R-L¹** and **R-L²** were obtained: 20025 and 20566 cm⁻¹ respectively. Among them, the energy gap (ΔE) between the excited singlet state S₁/S₁' and the excited triplet state of **R-L¹** and **R-L²** is both

greater than 5000 cm^{-1} , indicating that the intersystem crossing (ISC) process is effective. By reviewing the references (*Coord Chem Rev*, **2000**, *205*, 109–130; *J Am Chem Soc*, **1995**, *117*, 9408–9414; *Chem Eur J*, **2015**, *21*, 17921–17932; *Dyes Pigments*, **2022**, *206*, 110650–110662.), the energy value of the 5D_0 energy level of the Eu^{III} ion is 17500 cm^{-1} . Therefore, the energy differences between the excited triplet states of $\mathbf{R-L}^1$ and $\mathbf{R-L}^2$ and the 5D_0 of the Eu^{III} ion are approximately 2525 and 3066 cm^{-1} respectively. The above results prove that both $\mathbf{R-L}^1$ and $\mathbf{R-L}^2$ can efficiently sensitize the luminescence of Eu^{III} ions through the antenna effect. Relevant results have been added to Figure 4 in the “**Results**” section of the revised manuscript, and Figure 21 in the Supplementary Information.

The following description: “In order to deeply explore the energy transfer pathway of $\mathbf{R-Eu-Et-1/2}$, the Jablonski energy level diagram was further drawn (Fig. 4C). According to Reinhoudt’s rule of thumb^{21,48,49}, when the energy level difference between the excited singlet state and the excited triplet state of the ligand is greater than 5000 cm^{-1} , the intersystem crossing (ISC) process can be guaranteed to be effective. At room temperature, (1*R*/S,2*R*/S)-(+/-)-1,2-Diphenylethylenediamine and (1*R*/S,2*R*/S)-(+/-)-cyclohexanediamine were stirred with 1-Ethyl-2-imidazolecarboxaldehyde respectively, in 2 mL of CH_3OH solution for 24 h to obtain $\mathbf{R/S-L}^1$ and $\mathbf{R/S-L}^2$ (Supplementary Fig. 29). According to the solid-state UV-Vis absorption spectra of $\mathbf{R-L}^1$ and $\mathbf{R-L}^2$, it can be seen that the energy values of the excited singlet states S_1 and S_1' of $\mathbf{R-L}^1$ and $\mathbf{R-L}^2$ are 27174 and 28735 cm^{-1} respectively (Supplementary Fig. 30A, B). Obviously, the lowest excited state energy level of Gd^{III} ions (32500 cm^{-1}) is significantly higher than the excited singlet energy levels of $\mathbf{R-L}^1$ and $\mathbf{R-L}^2$, and energy cannot be transferred from the ligand to Gd^{III} ions. Therefore, the phosphorescence spectrum of $\mathbf{R-Gd-Et-1/2}$ was collected at 77 K, and the energy values of the excited triplet states T_1 and T_1' of $\mathbf{R-L}^1$ and $\mathbf{R-L}^2$ were obtained by Gaussian curve fitting to be 20025 and 20566 cm^{-1} , respectively (Supplementary Fig. 30C, D). The ΔE_L between the excited singlet state and excited triplet state of $\mathbf{R-L}^1$ and $\mathbf{R-L}^2$ was calculated to be 7149 and 8169 cm^{-1} respectively. Obviously, the above ΔE_L are all higher than 5000 cm^{-1} , indicating that $\mathbf{R-L}^1$ and $\mathbf{R-L}^2$ can effectively carry out the ISC process. The energy gap (ΔE) between the excited triplet energy level of the ligand and the lowest excited state energy level of Ln^{III} must be in a suitable range for effective energy transfer process^{50,51}. According to Latva’s

rule of thumb, a ΔE of 2000–5000 cm^{-1} is optimal for the energy transfer process of Eu^{III} , while the optimal ΔE of energy transfer for Tb^{III} ions is $2400 \pm 300 \text{ cm}^{-1}$ ⁵²⁻⁵⁴. In addition, the energy value of the 5D_0 energy level of Eu^{III} ions is 17500 cm^{-1} , and the energy value of the 5D_4 energy level of Tb^{III} is 20400 cm^{-1} . Further calculations show that the ΔE between the excited triplet states of $\mathbf{R-L}^1$ and $\mathbf{R-L}^2$ and the 5D_0 of Eu^{III} ions are 2525 and 3066 cm^{-1} respectively. Obviously, both $\mathbf{R-L}^1$ and $\mathbf{R-L}^2$ can efficiently sensitize the luminescence of Eu^{III} ions. In addition, the ΔE between the excited triplet states of $\mathbf{R-L}^1$ and $\mathbf{R-L}^2$ and the 5D_4 of Tb^{III} ions are -375 and 166 cm^{-1} respectively, indicating that they are unable to transfer energy to Tb^{III} ions through effective antenna effects (Fig. 4C).” has been added to section “**Solid State Luminescence and Circularly Polarized Luminescence of $R/S\text{-Eu-R-1}$ ($R = \text{Et/Me}$) and $R/S\text{-Eu-Et-2}$** ” of the revised manuscript and is highlighted in blue.

4) Consider moving the motivation for detecting Cu(II) to the introduction section to make the text more concise in the “Multiple Intelligent Sensing Properties” section.

Reply: Thank you very much for your suggestion. We have moved the description of the motivation for detecting Cu^{II} ions in the “**Multiple Smart Sensing Properties**” section to the “**Introduction**” section.

5) Clarify the specificity of the sensor to Cu(II) ions and its ability to distinguish from other species. Explain the horizontal axis on the limit of detection plot in Figure 4H.

Reply: Thank you very much for your suggestion. We have modified the horizontal axis in Figure 4H (Figure 5 in the revised manuscript) and have classified these references for Cu^{II} ion sensing.

6) Evaluate the dependence of sensor performance on pH values in the presence of Cu(II) ions and report the findings.

Reply: Thank you very much for your suggestion. We have supplemented the experimental results of optical sensing of Cu^{II} ions by $\mathbf{R-Eu-Et-1/2}$ under acidic or alkaline conditions, and obtained the LOD. Relevant results have been added to the “**Results**” section of the revised manuscript and Figures S39 and S40 in the Supplementary Information. Description below “In addition, the optical

response sensitivities of **R-Eu-Et-1/2** to Cu^{II} ions under different pH conditions were explored, and their LODs were obtained. Specifically, Cu^{II} ion aqueous solutions under acidic or alkaline conditions (pH = 4, 5, 6, 9, 10, and 11) were gradually added to solution **R-Eu-Et-1** or **R-Eu-Et-2** respectively, and it was observed that the characteristic emission of Eu^{III} ions of the above solutions gradually weakened and was eventually completely quenched. Through calculation, the LODs of **R-Eu-Et-1/2** for Cu^{II} ions under different pH conditions are 0.081 and 0.65 nM (pH = 4), 0.086 and 0.64 nM (pH = 5), 0.11 and 0.99 nM (pH = 6), 0.11 and 0.86 nM (pH = 9), 0.1 and 0.98 nM (pH = 10), 0.12 and 0.85 nM (pH = 11) (Supplementary Fig. 39, 40). The above experimental results prove that **R-Eu-Et-1/2** can be used to sense trace amounts of heavy metal ions Cu^{II} ions under different pH conditions, showing great potential for specific sensing of Cu^{II} ions in complex environmental pollutants.” has added the “**Results**” section of the revised manuscript and is highlighted in blue.

7) The figures’ quality in the PDF file provided for review is low and difficult to read. Higher-resolution figures should be included for clarity and ease of interpretation.

Reply: Thank you very much for your advice. We have increased the resolution of all images in the revised manuscript.

8) Reduce the use of unnecessary adjectives like “intelligent” or “smart” and instead use more precise descriptors.

Reply: Thank you very much for your suggestion. Based on your suggestions, we have carefully checked and deleted the relevant descriptions of “intelligent” or “smart” in the manuscript. In addition, we have carefully checked and corrected the language of the revised manuscript, and the manuscript has been carefully reviewed by an experienced editor whose first language is English and who specializes in editing papers written by scientists whose native language is not English.

9) Clarify the meaning of the horizontal axis in Figure 5G.

Reply: Thank you very much for your suggestion. We have modified the horizontal axis in Figure 5G (Figure 6G in the revised manuscript) and have classified these reported references of responses to acids or bases.

10) Explain the main advantages of the proposed anti-counterfeiting approach compared to other existing technologies. Make the wetting of the sensor with a Cu(II)-containing solution clearer. Maintain consistent nomenclature as Cu(II) and remove Cu²⁺ from the figure.

Reply: Thank you for your suggestion. The existing anti-counterfeiting technologies mainly include laser anti-counterfeiting, digital anti-counterfeiting, texture anti-counterfeiting, QR code anti-counterfeiting, and chip anti-counterfeiting. These technologies usually use special precision equipment such as strong light, ultraviolet light, and infrared light or electronic communication equipment for anti-counterfeiting identification. In addition, the above-mentioned anti-counterfeiting technologies are usually single-layer anti-counterfeiting phenomena. However, the anti-counterfeiting methods mentioned in this manuscript are relatively low-cost, simple in design, and do not require specialized equipment or operation by professionals. In addition, fluorescent ink anti-counterfeiting technology has been widely used due to its advantages of clarity, speed and high degree of visualization. Finally, we have maintained a consistent writing format for the names of Cu^{II} ions in the manuscript.

11) Address whether the sensor's response is reversible after coming into contact with Cu(II) ions in the “Double Anti-counterfeiting Performance” section.

Reply: Thank you very much for your suggestion. We believe that **R-Et-Eu-1** is irreversible after sensing Cu^{II} ions. This is mainly because Cu^{II} ions replace Eu^{III} ions to form a more stable complex. To verify our idea, the following experiments were designed: the fragments in **R-Et-Eu-1**+Cu^{II} solution were monitored for their stability under HRESI-MS conditions with different ion source energies (Figure R1). When the ion source energy of HRESI-MS was 0 eV, four molecular ion peaks were captured: [Cu(**R-L**¹)]⁺, [Cu(**R-L**¹)(NO₃)]⁺, [Eu(**R-L**¹)(NO₃)₂]⁺, and [Eu(**R-L**¹)(NO₃)₂(DMF)]⁺ proving the formation of Cu^{II} complexes. As the ion source energy of HRESI-MS continues to increase, [Cu(**R-L**¹)]⁺ and [Cu(**R-L**¹)(NO₃)]⁺ still maintain high stability,

while the $[\text{Eu}(\mathbf{R-L}^1)(\text{NO}_3)_2]^+$ and $[\text{Eu}(\mathbf{R-L}^1)(\text{NO}_3)_2(\text{DMF})]^+$ fragments obviously disappear, indicating that the Cu^{II} complex has higher stability. Therefore, we believe that the response of $\mathbf{R-Et-Eu-1}$ after contact with Cu^{II} ions is irreversible.

Figure R1. Simulation results of HRESI-MS spectra and fragments in a mixed solution of $\mathbf{R-Et-Eu-1} + \text{Cu}^{\text{II}}$ bombarded with different ion source energies.

REVIEWER COMMENTS

Reviewer #1 (Remarks to the Author):

The authors have made extensive modifications to the previous version of the manuscript. However, they did not solve the core concern of the manuscript but rather exposed a series of fatal flaws. Consequently, this article does not meet the standards for publishing in Nature Communications. Some fatal flaws that exist in the manuscript are listed below:

(1) The data in this article contains obvious inconsistencies. For example, the author claims “to have observed the R-Eu-Et-1/2 were still observed to have instantaneous bright solid-state luminescence when the 365 nm UV-light was turned off (Supplementary Fig. 26)”. The author took a series of photos to demonstrate the luminescence of the complex after turning off the 365 nm UV-light. However, the reviewer believes that this dataset is inconsistent with the facts because it conflicts significantly with the solid-state luminescence lifetime data presented later (Fig. 4B and Supplementary Fig. 27). If the solid complexes indeed exhibits the phenomenon of luminescence persistence after turning off the UV lamp, this phenomenon will be reflected in the test of luminescence lifetime, manifested as an extremely long decay curve in the luminescence lifetime test and a long luminescence lifetime matching the duration of the luminescence persistence phenomenon. But the two sets of data provided by the author have a difference of 1000 times in time scale (0.83s & 930/923, 980/975 μ s), which is extremely absurd. Why does the photograph taken by the author show that after the 365 nm UV-light is turned off, there is still background light except for the luminescence of the complexes? Does the author manipulate the data artificially?

(2) The testing procedures in this manuscript are not standardized, which results in a lack of reliability in many of its findings. For example, in the author's manuscript, a EuIII complex Eu-Et-3 was successfully obtained which does not have a rotor and vibrational primitives and has the same coordination environment as R-Et-Eu-1. However, in the subsequent AIE tests, complex R-Et-Eu-1 and complex Eu-Et-3 were tested using different solvent conditions (glycerol/DMSO and water/DMF). The different testing conditions rendered the test results of complex R-Et-Eu-1 and complex Eu-Et-3 meaningless for comparison. In addition, in the subsequent sensing tests, are the test conditions for Complex R-Et-Eu-1/2 and Complex Eu-Et-3 completely consistent? The author only mentioned that the testing condition for Complex R-Et-Eu-1/2 is an acetonitrile solution, but did not specify the testing solvent for Complex Eu-Et-3 in the manuscript or the Supplementary Fig. 34. If the test conditions are different, comparing LOD would be meaningless.

(3) The reviewer insists that the irreversible destruction of the complex structure to achieve a weakening in luminescence is not a valuable sensing approach. By this way, any substrates that can destroy the complex's structure all will response to this luminescent quenching. The newly added acid-base sensing part by the author is still achieved through the degradation of the complex to achieve luminescence weakened.

Reviewer #2 (Remarks to the Author):

Zou et al. used chiral ligands with molecular rotor structures or vibration units to synthesize three pairs of dynamic chiral EuIII complex emitters with aggregation-induced enhanced antenna effects, and demonstrated the AIE behavior of these complexes. In the revised manuscript, the author combined DTF theoretical calculations, photophysical parameters, and many references to clarify the energy transfer pathway of the EuIII complex emitters. The author added many experimental results such as DLS, Zeta potential, AIE titration curve, and sensing experiments on CuII, CoII, and FeIII ions under different conditions. In addition, the author summarized a series of work, compared them, proved that the BCPL value of R/S-Eu-Et-1 was at a high level and achieved a rare double improvement of glum and BCPL. Overall, the revised manuscript is well structured and of good writing quality. The pictures are clearly presented. After carefully reading the revised manuscript and the author's response letter, I have come to the following conclusions. In both the revised manuscript and SI, the authors followed the suggestions of the editor and all reviewers to provide additional experimental data or further evaluation of the data to support the claims of the manuscript. Based on these observations, I think the author has answered all my concerns in a professional and accurate manner and responded to all comments satisfactorily. The manuscript has improved greatly. The novel points have been identified, and the revised manuscript more fully elaborates and validates the claims of the original manuscript. Given this revision, I strongly support to accept this manuscript for publication in Nature Communications.

Reviewer #3 (Remarks to the Author):

The authors appropriately addressed the majority of the raised questions and the manuscript was significantly improved. Nevertheless, the following minor issues deserve further revision:

1. In the sentence "(...) It is worth noting that the solid-state luminescence QYs of R-Eu-Et-1/2 with rotors or vibration units are much higher than that of Eu-Et-3, where the QY of R-Eu-Et-1/2 are 9.76 and 8.57 times that of Eu-Et-3, respectively" (Page 6), there is no rationale to use two decimal figures.
2. The majority of the measured values reported in the manuscript do not have errors. This must be corrected. For instance, add the errors to the lifetime values and energies of the 5D0 level and excited singlet and triplet states (pages 5, 7, and 8), as well as the difference in energy between them.
3. Figure 4B displays the decay curves of the 5D0 level in the R-Eu-Et-1/2 compounds in the solid state and not the "solid-state luminescence lifetimes of R-Eu-Et-1/2" as written in the caption. Lifetime is an intrinsic characteristic of a particular energy level.
4. Fig. 6 is too small and difficult to read.

Responses to Reviewers:

Reviewer 1:

The authors have made extensive modifications to the previous version of the manuscript. However, they did not solve the core concern of the manuscript but rather exposed a series of fatal flaws. Consequently, this article does not meet the standards for publishing in *Nature Communications*.

Reply: Thank you very much for taking time out of your busy schedules to review the manuscript and provide suggestions. In the previous round of review comments, we effectively revised the series of questions you raised and made our best efforts to resolve the doubts of all reviewers and editors. Specifically, we synthesized **Eu-Et-3** without a rotor and vibration unit and with ACQ behavior. Photophysical studies have shown that the introduction of molecular rotors or vibration units can not only effectively improve the luminescence properties of lanthanide complexes, but also greatly improve their optical sensing sensitivity to copper ions. By synthesizing Gd(III) and Tb(III) complexes that do not match the energy levels of L^1 and L^2 , their emission spectra in mixed solutions of glycerol/DMSO at different ratios were tested. The results show that these lanthanide complexes have significant AIE behavior, and it is not the change in luminescence caused by the reduction of the T_1 state of the organic ligand. In addition, the absorption peaks of these lanthanide complexes in the molecular state and aggregation state respectively show a certain degree of red shift, which is caused by changes in their sizes in the two different states. At the same time, the DLS results also prove this statement. The quantum yields of **R/S-Eu-Et-1/2** and **R/S-Eu-Me-1** in mixed solutions with different proportions were further supplemented. Experimental results show that as the proportion of glycerol or acetonitrile continues to increase, the quantum yields of these lanthanide complexes continue to increase, once again proving that they have significant AIE behavior. Finally, the energy transfer path depicted in the Jablonski energy level diagram is detailed through the calculation of energy values. Overall, we responded effectively to all reviewers' and editor's doubts and comments and made efforts to produce significant revisions. In addition, we have carefully processed each comment from all reviewers and editors in the second round of review. Obviously, we believe that the revised manuscript has outstanding highlights, novel content, clear logic, and reliable results. Therefore, after two rounds of revisions, the quality of the manuscript has been greatly improved, and we believe that the revised manuscript can meet the publication standards of *Nature Communications*.

Some fatal flaws that exist in the manuscript are listed below:

(1) The data in this article contains obvious inconsistencies. For example, the author claims “to have observed the **R-Eu-Et-1/2** were still observed to have instantaneous bright solid- state luminescence when the 365 nm UV-light was turned off (Supplementary Fig. 26)”. The author took a series of photos to demonstrate the luminescence of the complex after turning off the 365 nm UV-light. However, the reviewer believes that this dataset is inconsistent with the facts because it conflicts significantly with the solid-state luminescence lifetime data presented later (Fig. 4B and Supplementary Fig. 27). If the solid complexes indeed exhibit the phenomenon of luminescence persistence after turning off the UV lamp, this phenomenon will be reflected in the test of luminescence lifetime, manifested as an extremely long decay curve in the luminescence lifetime test and a long luminescence lifetime matching the duration of the luminescence persistence phenomenon. But the two sets of data provided by the author have a difference of 1000 times in time scale (0.83s & 930/923, 980/975 μ s), which is extremely absurd. Why does the photograph taken by the author show that after the 365 nm UV-light is turned off, there is still background light except for the luminescence of the complexes? Does the author manipulate the data artificially?

Reply: Thank you for your suggestion. You think: “If the solid complexes indeed exhibits the phenomenon of luminescence persistence after turning off the UV lamp, this phenomenon will be reflected in the test of luminescence lifetime, manifested as an extremely long decay curve in the luminescence lifetime test and a long luminescence lifetime matching the duration of the luminescence persistence phenomenon.”. However, we think that you misunderstood the fluorescence lifetime of the molecule. The fluorescence lifetime of a molecule is the average length of time that photons are in the excited state. For example, the standard definition of fluorescence lifetime of a molecules can be obtained from the following authoritative books and web pages:

- [1] <https://www.horiba.com/int/scientific/technologies/fluorescence-spectroscopy/principles-and-theory-of-fluorescence-spectroscopy/>
- [2] <https://www.edinst.com/blog/what-is-fluorescence-lifetime/>
- [3] *J. Breen, et al. CHM 331 Advanced Analytical Chemistry 1, 2022, pp 4–517.*
- [4] *H. Sahoo, et al. Optical Spectroscopic and Microscopic Techniques: Analysis of Biological Molecules, 2022, pp 1–260.*

- [5] B. Valeur, M. N. Berberan-Santos, *Molecular Fluorescence: Principles and Applications*. Wiley-VCH Verlag GmbH & Co. KGaA, 2012, pp 1–569.
- [6] J. R. Lakowicz, *et al. Principles of Fluorescence Spectroscopy*. 3rd edition, Springer US, 2006, pp 1–673.
- [7] D. M. Jameson, *et al. Introduction to Fluorescence*. CRC Press, 2014, pp 1–26.

However, the photographing time provided in Supplementary Figure 26 refers to the time when the photon in the excited state is completely decayed, which can be reflected in the time it takes for the photon to almost return to the ground state completely on the decay curve. Therefore, the time given in Fig. 4B of the revised manuscript and Supplementary Figure 26 are not the same concept as you think. Furthermore, the way the decay curves for ***R/S-Eu-R-1*** (**R = Et/Me**), ***R/S-Eu-Et-2***, and their dispersion in an acetonitrile solution are plotted in the revised manuscript (Fig. 4B, Supplementary Figure 19 and 27) do not show the time required for the photon to completely decay. Subsequently, we further transformed the above decay curve spectrum and found that within our test interval, the excited state photons of these lanthanide complex emitters did not completely decay back to the ground state (Figure R1). Therefore, there is no correspondence between the photographing time provided in Supplementary Figure 26 and the time obtained through the decay curve in Fig. 4B of the revised manuscript. In addition, we extended the test time and further retested the fluorescence lifetime of ***R/S-Eu-R-1*** (**R = Et/Me**) and ***R/S-Eu-Et-2*** and dispersed them in acetonitrile solution. And the time picture of the complete decay of photons in the excited state of these lanthanide complex emitters was obtained (Figure R2). Experimental results show that the fluorescence lifetimes of ***R/S-Eu-R-1*** (**R = Et/Me**), ***R/S-Eu-Et-2***, and their dispersion in acetonitrile solution are: the lifetimes of ***R/S-Eu-R-1*** (**R = Et/Me**) and ***R/S-Eu-Et-2*** were $\tau_{1/e} = 934.4 (\pm 0.009)/929.5 (\pm 0.008)$, $979.3 (\pm 0.009)/977.0 (\pm 0.008)$ and $1112.9 (\pm 0.021)/1039.8 (\pm 0.009)$ μs , respectively (Fig. 4B and Supplementary Fig. 27) and the lifetimes of ***R/S-Eu-R-1*** (**R = Et/Me**) and ***R/S-Eu-Et-2*** dispersed in CH_3CN were $\tau_{1/e} = 1207.5 (\pm 0.022)$, $1225.9 (\pm 0.027)$, $1223.6 (\pm 0.031)$, $1246.4 (\pm 0.028)$, $1240.8 (\pm 0.047)$, and $1237.5 (\pm 0.023)$ μs , respectively (Supplementary Figure 19). Obviously, the lifetimes of these compounds obtained after extending the test time of photon decay are almost consistent with the results of previous tests, proving the reproducibility of the test results. More importantly, it can be obtained from the decay curve after extending the photon decay test time that the complete decay time of photons of ***R/S-Eu-R-1*** (**R = Et/Me**) and

R/S-Eu-Et-2 were 12.93/13.82 ms, 13.79/13.74 ms, and 13.31/14.11 ms respectively, and the time for complete decay of photons of *R/S-Eu-R-1* ($R = \text{Et/Me}$) and *R/S-Eu-Et-2* dispersed in CH_3CN were 15.49/15.29 ms, 15.53/15.74 ms, and 14.01/14.03 ms, respectively.

Figure R1. The decay curves of 5D_0 energy level in solid *R/S-Eu-R-1* ($R = \text{Eu/Me}$) and *R/S-Eu-Et-2* in the Revised Manuscript (R1).

Figure R2. The decay curves of 5D_0 energy level in solid *R/S-Eu-R-1* ($R = \text{Eu/Me}$) and *R/S-Eu-Et-2* obtained by extending the test time.

In addition, regarding the photos and the time shown in Supplementary Figure 26, we used the video function in the original camera that comes with the iPhone system to record the video of

R-Eu-Et-1/2. Then we used the phone's built-in video editing function to split the original video into photos. A total of 10 pictures of **R-Eu-Et-1** and 8 pictures of **R-Eu-Et-2** were extracted from the video taken from the moment the UV lamp was turned off. Because the resolution of this video software is only 12 frames per second. Therefore, the calculation of the time points of the photos split by this video software is: $(1/12)*10=0.833$ s (**R-Eu-Et-1**) and $(1/12)*8=0.664$ s (**R-Eu-Et-2**). However, upon further inquiry, we found that the video per second was not just the highest to 12 frames of photos, indicating that the video software has a low resolution for splitting the photos in the video. We imported the original video into a more professional video processing software (Adobe Premiere Pro 2022, videos can be played at slow speed and then split into 24 frames, this software has very high resolution for extracting images from videos) to more accurately correct the time that photons of **R-Eu-Et-1/2** completely decay in the excited state. Specifically, first, the original video of **R-Eu-Et-1/2** was imported into Adobe Premiere Pro 2022 video software and the original video was played ten times slower. From the moment the UV lamp was turned off, 10 frames of photos of each of **R-Eu-Et-1/2** were captured (Figure R3). After calculation, it can be found that the time of **R-Eu-Et-1/2** is approximately $(10*(1/24))/10 = 0.0416$ s (41.6 ms) and $(10*(1/24))/10 = 0.0416$ s (41.6 ms) respectively. Obviously, the time for complete decay of photons in the excited state of these lanthanide complex emitters (**R/S-Eu-Et-1**: 12.93 ms, and **R/S-Eu-Et-2**: 13.31 ms) is very close to the results observed in Adobe Premiere Pro 2022 video software. In addition, we screened a large number of references and found that reports from many well-known research groups also showed that there is a certain difference between the lifetime of the compound and the time for the photons in the excited state to completely decay. For example:

Lifetime	Lasting luminescent time after UV-off	References
1506 ms	13 s	F. Peng, et al. J. Am. Chem. Soc. , 2023, DOI:10.1021/jacs.3c07034.
0.83 s, 0.69 s, and 0.79 s	20 s, 5 s, and 15 s	X. Liu, et al. Adv. Funct. Mater. , 2023, 2310198. DOI: 10.1002/adfm.202310198.
2.7 s	20 s	Z. An, et al. Nat. Commun. , 2022, 13 , 4890.
0.8 s and 3.2 s	5 s and 25 s	M. Wagner, et al. Angew. Chem. Int. Ed. , 2023, 62 , e202215071.
1.65 s, 1.24 s, 1.39 s, and 0.11s	15 s, 9 s, 9 s, and 7s	B. Z. Tang, et al. Adv. Funct. Mater. , 2023, 2312883. DOI: 10.1002/adfm.202312883.
30 ms	6 s	Y. Zhao, et al. J. Am. Chem. Soc. , 2021, 143 , 18527–18535.
540 ms	3 s	T. Chen, et al. Adv. Funct. Mater. , 2023, 2310043. DOI: 10.1002/adfm.202310043.

20 ns	10 s	W. Huang, et al. Nat. Commun. , 2020, 11 , 4802.
98.5 ms	12 min	B. Z. Tang, et al. J. Am. Chem. Soc. , 2022, 144 , 3050–3062.
109 ms	4 s	D. Yan, et al. Adv. Mater. , 2021, 33 , 2007571.
309 ms	10 s	K. Zhang, et al. Angew. Chem. Int. Ed. , 2021, 60 , 17138–17147.
1.75 ms	7 s	J. Zhuang, et al. Angew. Chem. Int. Ed. , 2021, 60 , 22253–22259.
1595.77 μ s	60 s	W. Liu, et al. Chem. Eng. J. , 2023, 476 , 146487.
422 ms	13 s	D. F. Lu, et al. Chem. Eng. J. , 2024, 479 , 147851.

It is worth noting that most of the decay times of lanthanide complex emitters are in the millisecond range, and most are less than 100 ms. However, due to the resolution limitations of current video and shooting software, it is difficult to capture the instant afterglow phenomenon of lanthanide complex emitters after the UV lamp is turned off. Therefore, none of the research works on the optical properties of lanthanide complex emitters provide pictures of the afterglow after the UV lamp is turned off. Here, the original intention of providing Supplementary Figure 26 is to allow readers to understand and observe the afterglow phenomenon of lanthanide complex emitters more intuitively. Therefore, we would like to emphasize again that there is no data manipulation and artificial modification time with respect to Supplementary Figure 26.

More importantly, regarding the background light issue of Supplementary Figure 26. This is mainly due to the fact that after the UV lamp is turned off, the lanthanide complex still has a strong afterglow, causing the outline of ordinary filter paper to appear. If there is still external UV light irradiation (for example, when the UV lamp is turned on) after the UV lamp is turned off, the glass sheet under the filter paper will have a significant reflection effect on the UV lamp. However, we don't see any reflective effect on the glass pieces in the picture. Obviously, there is no external UV lamp irradiating **R-Eu-Et-1/2** during the entire process after the UV lamp is turned off. In addition, we have uploaded the original video and slow-play video of **R-Eu-Et-1/2** to the submission system.

Figure R3. After turning off the 365 nm UV-light, ***R-Eu-Et-1*** (A) and ***R-Eu-Et-2*** (B) can still keep red light for a short time (play the original video of ***R-Eu-Et-1/2*** ten times slower through Adobe Premiere Pro 2022 video software, and from the moment the UV lamp was turned off, 10 frames of each of ***R-Eu-Et-1/2*** were captured).

(2) The testing procedures in this manuscript are not standardized, which results in a lack of reliability in many of its findings. For example, in the author's manuscript, a Eu^{III} complex ***Eu-Et-3*** was successfully obtained which does not have a rotor and vibrational primitives and has the same coordination environment as ***R-Et-Eu-1***. However, in the subsequent AIE tests, complex ***R-Et-Eu-1*** and complex ***Eu-Et-3*** were tested using different solvent conditions (glycerol/DMSO and water/DMF). The different testing conditions rendered the test results of complex ***R-Et-Eu-1*** and complex ***Eu-Et-3*** meaningless for comparison. In addition, in the subsequent sensing tests, are the test conditions for complex ***R-Et-Eu-1/2*** and complex ***Eu-Et-3*** completely consistent? The author only mentioned that the testing condition for complex ***R-Et-Eu-1/2*** is an acetonitrile solution, but did not specify the testing solvent for complex ***Eu-Et-3*** in the manuscript or the Supplementary Fig. 34. If the test conditions are different, comparing LOD would be meaningless.

Reply: Thank you for your suggestion. Based on your suggestion, we additionally tested the

emission spectra of **Eu-Et-3** dissolved in acetonitrile/DMF mixed solutions (acetonitrile is a poor solvent) with different acetonitrile contents (f_w). Experimental results show that as the proportion of acetonitrile increases, the emission intensity of **Eu-Et-3** rapidly weakens, indicating that it has an obvious ACQ effect. Therefore, the above experimental results and the experimental results of the revised manuscript (R1) both indicate that **Eu-Et-3** has an obvious ACQ effect. And the following description: “Using acetonitrile as a poor solvent, **Eu-Et-3** was dissolved in acetonitrile/DMF mixed solutions with different acetonitrile (f_w) contents, and the emission spectra of the above solutions were tested. The results show that as the proportion of acetonitrile increases, the emission intensity of **Eu-Et-3** rapidly weakens, proving that it has an obvious ACQ effect.” has been added to the revised manuscript and Supplementary Information and highlighted in blue. In addition, the test solvents, and conditions for the sensing experiments of complexes **R-Et-Eu-1/2** and **Eu-Et-3** in the revised manuscript are completely consistent. We have added these specific details and related information to the revised manuscript and supporting information.

(3) The reviewer insists that the irreversible destruction of the complex structure to achieve a weakening in luminescence is not a valuable sensing approach. By this way, any substrates that can destroy the complex’s structure all will response to this luminescent quenching. The newly added acid-base sensing part by the author is still achieved through the degradation of the complex to achieve luminescence weakened.

Reply: Thank you for your suggestion. You insist that: “the irreversible destruction of the complex structure to achieve a weakening in luminescence is not a valuable sensing approach. By this way, any substrates that can destroy the complex’s structure all will response to this luminescent quenching.” However, we do not agree with you. To date, many well-known research groups and scientists have used the design principles of crystal engineering and structure-property correlation to develop complexes with a wide range of fluorescence-sensing properties and explore their various sensing mechanisms (the main sensing mechanisms include: radiative energy transfer, competitive absorption, and FRET. Among them, both radiative energy transfer and competitive absorption will lead to obvious emission quenching; the FRET process may lead to emission quenching of the probe material, emission enhancement of the probe material, or induction of fluorescence in the analyte species). This work has facilitated the development of many key sensing areas, including

biomolecules, environmental toxins, explosives, ionic substances, and more. Notably, most sensing mechanisms are associated with significant quenching of the probe. In addition, the following advantages should be sought when developing any chemical sensor, such as high sensitivity, good selectivity, fast response time, and high stability. Generally, only very extreme conditions will affect the performance of chemical sensors. However, realistic sensing conditions are not extreme. Furthermore, low sensitivity, long response times, and limited target analytes increase their cost. Not only that, if ion sensors are to have broad applicability, detection selectivity is extremely important, even in the presence of many other potentially interfering anions. And while the parameters controlling sensitivity and selectivity are being explored in the experimental phase, it remains an area that requires significant progress to reach performance levels for practical applications. Therefore, it is urgent to explore and develop chemical sensors with high sensitivity, short response time, high selectivity and emit strong light in the open state. Although the structure of **R-Eu-Et-1/2** was damaged during the sensing process. Importantly, **R-Eu-Et-1/2** has extremely high sensitivity (LODs as low as 2.55 and 4.44 nM, respectively), fast response time, excellent selectivity, and anti-interference ability. More importantly, in the metal sensing part of the revised manuscript, only the sensed metal ions will destroy the structure of **R-Eu-Et-1/2**, resulting in emission quenching of the lanthanide complex. However, other control metal ions will not destroy the structure of **R-Eu-Et-1/2** and cause the emission quenching of the complex. Therefore, we believe that the research on “intelligent sensing of heavy metal ions” in the manuscript is of great significance. Not only that, but we also screened a large number of references and found that many well-known research groups have published research on “achieving luminescence quenching or turning on by destroying the structure of the complex” in high-level journals. For example:

1. J. Li, *et al.* Metal–organic frameworks: functional luminescent and photonic materials for sensing applications, *Chem. Soc. Rev.*, **2017**, *46*, 3242–3285.
2. B. Zou, *et al.* Stimuli-Responsive Luminescent Properties of Tetraphenylethene-Based Strontium and Cobalt Metal–Organic Frameworks, *Angew. Chem. Int. Ed.* **2020**, *59*, 19716–19721.
3. C. Duan, *et al.* Multifunctional Mesoporous Silica Material Used for Detection and Adsorption of Cu²⁺ in Aqueous Solution and Biological Applications *in vitro* and *in vivo*, *Adv. Funct. Mater.* **2010**, *20*, 1903–1909.

4. B. Wang, *et al.* Tuning the Luminescence of Metal–Organic Frameworks for Detection of Energetic Heterocyclic Compounds, *J. Am. Chem. Soc.*, **2014**, *136*, 15485–15488.
5. M. Oh, *et al.* Dual Changes in Conformation and Optical Properties of Fluorophores within a Metal–Organic Framework during Framework Construction and Associated Sensing Event, *J. Am. Chem. Soc.*, **2014**, *136*, 12201–12204.

In summary, we think that **R-Eu-Et-1/2**, which is synthesized only through a simple, efficient, convenient, and high-atom utilization solvothermal “one-pot” method, shows obvious advantages over traditional luminescent materials in many aspects. And its extremely high sensitivity and fast response time are expected to promote the development of advanced chemical sensors. Finally, we hope that all the above responses clearly convey the novelty and plausibility of our work and demonstrate its suitability for publication in *Nature Communications*.

Reviewer 2:

Zou *et al.* used chiral ligands with molecular rotor structures or vibration units to synthesize three pairs of dynamic chiral Eu^{III} complex emitters with aggregation-induced enhanced antenna effects, and demonstrated the AIE behavior of these complexes. In the revised manuscript, the author combined DFT theoretical calculations, photophysical parameters, and many references to clarify the energy transfer pathway of the Eu^{III} complex emitters. The author added many experimental results such as DLS, Zeta potential, AIE titration curve, and sensing experiments on Cu^{II}, Co^{II}, and Fe^{III} ions under different conditions. In addition, the author summarized a series of work, compared them, proved that the B_{CPL} value of **R/S-Eu-Et-1** was at a high level and achieved a rare double improvement of g_{lum} and B_{CPL} . Overall, the revised manuscript is well structured and of good writing quality. The pictures are clearly presented. After carefully reading the revised manuscript and the author’s response letter, I have come to the following conclusions. In both the revised manuscript and SI, the authors followed the suggestions of the editor and all reviewers to provide additional experimental data or further evaluation of the data to support the claims of the manuscript. Based on these observations, I think the author has answered all my concerns in a professional and accurate manner and responded to all comments satisfactorily. The manuscript has improved greatly. The novel points have been identified, and the revised manuscript more fully elaborates and validates the claims of the original manuscript. Given this revision, I strongly

support to accept this manuscript for publication in *Nature Communications*.

Reply: Thank you very much for your careful review of the manuscript. Your professional guidance and patient answers have greatly improved the quality of the manuscript and benefited us a lot. Thank you again for your affirmation and high praise of our work.

Reviewer 3:

The authors appropriately addressed the majority of the raised questions and the manuscript was significantly improved. Nevertheless, the following minor issues deserve further revision:

Reply: Thank you very much for spending your valuable time and expertise to evaluate our work. We are very happy and honored to receive your positive comments and have made great efforts to resolve your confusion. Your comments greatly enriched the content and highlights of the manuscript, which greatly improved the quality of the manuscript.

1. In the sentence “(...) It is worth noting that the solid-state luminescence QYs of **R-Eu-Et-1/2** with rotors or vibration units are much higher than that of **Eu-Et-3**, where the QY of **R-Eu-Et-1/2** are 9.76 and 8.57 times that of **Eu-Et-3**, respectively” (Page 6), there is no rational to use two decimal figures.

Reply: Thank you very much for your comment. Based on your suggestion, we have changed “where the QY of **R-Eu-Et-1/2** is 9.76 and 8.57 times that of **Eu-Et-3**, respectively” to “where the QY of **R-Eu-Et-1/2** are about 10 and 9 times that of **Eu-Et-3**, respectively.” in the revised manuscript.

2. The majority of the measured values reported in the manuscript do not have errors. This must be corrected. For instance, add the errors to the lifetime values and energies of the 5D_0 level and excited singlet and triplet states (pages 5, 7, and 8), as well as the difference in energy between them.

Reply: We would like to thank you for pointing this out, we have added errors to the revised manuscript for the 5D_0 energy level and the lifetime values and energies of the excited singlet and triplet states, as well as the energy difference between them.

3. Figure 4B displays the decay curves of the 5D_0 level in the **R-Eu-Et-1/2** compounds in the solid state and not the “solid-state luminescence lifetimes of **R-Eu-Et-1/2**” as written in the caption. Lifetime is an intrinsic characteristic of a particular energy level.

Reply: Thank you very much for reminding. We have corrected the relevant figure captions in the revised manuscript. We have changed “Solid-state luminescence lifetimes of **R-Eu-Et-1/2**” to “The decay curve of 5D_0 energy level in solid-state **R-Eu-Et-1/2**”; changed “Luminescence lifetimes of **R/S-Eu-R-1** (**R = Et/Me**) (A–D) and **R/S-Eu-Et-2** (E and F) dispersed in CH₃CN.” to “The decay curve of 5D_0 energy level of **R/S-Eu-R-1** (**R = Et/Me**) (A–D) and **R/S-Eu-Et-2** (E and F) dispersed in CH₃CN.”; changed “Luminescence lifetimes of **S-Eu-Et-1**, **R/S-Eu-Me-1**, and **S-Eu-Et-2** in aggregated state.” to “The decay curve of 5D_0 energy level in solid-state **S-Eu-Et-1**, **R/S-Eu-Me-1**, and **S-Eu-Et-2**.”; and changed “Luminescence lifetime of **Eu-Et-3** in aggregated state.” to “The decay curve of 5D_0 energy level in solid-state **Eu-Et-3**.”.

4. Fig. 6 is too small and difficult to read.

Reply: Thank you for your suggestion, we have refined Figure 6 and adjusted its clarity.

REVIEWERS' COMMENTS

Reviewer #1 (Remarks to the Author):

I satisfy the modifications of authors to modified manuscript, and agree to its acceptance by NC